# GREC: DOUBLY EFFICIENT PRIVACY-PRESERVING RECOMMENDER SYSTEMS FOR RESOURCE-CONSTRAINED DEVICES

## ABSTRACT

Federated recommender system (FedRec) has emerged as a solution to protect user data through collaborative training techniques. However, the real-world implementation of FedRec is hindered by two critical resource constraints of edge devices: a) limited upload bandwidth and b) limited user computational power and storage. Existing methods addressing the first issue, such as message compression techniques, often result in accuracy degradation or potential privacy leakage. For the second issue, most federated learning (FL) protocols assume that users must store and maintain the entire model locally for private inference, which is resource intensive. To address these challenges, we propose doubly efficient privacy-perserving recommender systems (GREC) consisting of both training and inference phase. To reduce communication costs during the training phase, we design a lossless secure aggregation (SecAgg) protocol based on functional secret sharing leveraging the sparsity of the update matrix. During the inference phase, we implement a user-side post-processing local differential privacy (LDP) algorithm to ensure privacy while shifting the bulk of computation to the cloud. Our framework reduces uplink communication costs by up to 90x compared to existing SecAgg protocols and decreases user-side computation time during inference by an average of 11x compared to full-model inference. This makes GREC a practical and scalable solution for deploying federated recommender systems on resource-constrained devices.

## 1 INTRODUCTION

Personalized recommendation systems (RecSys) model the interactions between users and items to uncover users' interests. To understand the underlying preferences of users and properties of items, various model-based approaches have been developed to learn hidden representations of both users and items Koren et al. (2009); Xue et al. (2017); Rendle (2010). These methods embed users and items into fixed-size latent vectors, which are then used to predict interactions. The parameters of these latent vectors are known as user and item embeddings.

The development of personalized recommendation systems (RecSys) relies heavily on collecting user profiles and behavioral data, such as gender, age, and item interactions. However, the sensitive nature of this information often makes users hesitate to share it with service providers. Recent advancements in edge computing have provided a potential solution through federated learning (FL), which allows users to collaboratively train models on their local devices without exposing personal data McMahan et al. (2017). In a typical FL setup, a central server aggregates model updates from multiple edge devices. The aggregated parameters are then redistributed to the devices for further training. During inference, users employ the locally stored global model to make predictions on previously unseen items, maintaining privacy while benefiting from collective learning.

Despite various FL-based architectures proposed for RecSys Ammad-Ud-Din et al. (2019); Chai et al. (2020); Jia & Lei (2021); Wang et al. (2022), there remains a significant gap between research and real-world applications, mainly due to the resource limitations of edge devices. Two critical resource constraints in FL-based RecSys are: (1) *Limited upload bandwidth.* Clients in federated networks often experience slower upload speeds compared to downloads. This issue becomes particularly

problematic in latent factor-based RecSys, where the communication payload for uploads scales linearly with the number of items, potentially leading to substantial transmission overhead in practical applications. (2) *Limited user computational power and storage.* Edge devices generally have limited processing capabilities, memory, and storage compared to centralized servers. Running large models locally can place a heavy computational burden on users, degrading performance and negatively impacting the user experience.

To address the first constraint, existing message compression methods can be categorized into three categories: (1) Top-K sparsification that transmits a subset of updates to reduce the message size Gupta et al. (2021); Aji & Heafield (2017). (2) Quantization that represents each element in the gradients with fewer bits Zheng et al. (2023). (3) Dimension reduction that involves factorizing the transmission matrix into low-rank matrices or projecting the matrix into a lower-dimensional space Nguyen et al. (2024); Shin et al. (2018). However, these methods are lossy in principle and often result in non-negligible accuracy loss in practice, highlighting the need for a lossless communication-efficient approach. For the second constraint, many FL protocols require users to store the entire global model for both training and inference, placing considerable demands on long-term storage and computational resources—especially for resource-constrained edge devices.

In practice, users typically interact with only a small subset of available items, which presents two key opportunities for payload optimization. First, during training, only the embeddings of the items a user interacts with are relevant for model updates. As a result, users can store and update only these relevant item embeddings, significantly reducing computational and storage overhead. While this approach effectively addresses the training phase, there is a noticeable lack of research focused on minimizing costs during inference. For subsequent recommendations, users are often required to retain the entire item embedding matrix, which is resource-intensive. An ideal solution would enable cloud-based inference that preserves user privacy by keeping their embeddings and features secure from the server, while offloading the bulk of the computational burden.

Secondly, the update vector is highly sparse in that only a small subset of the item embeddings are non-zero. Therefore, it is desirable to make the per-user communication succinct, i.e., independent of or logarithmic in the item size. A naive solution is to transmit only updates of rated items to the server, possibly supplemented with fake items sampled Lin et al. (2022). However, this method compromises privacy by revealing which items the user has rated, or significantly narrowing the set of potential rated items. To protect user interaction data, it is crucial to ensure that no information about the specific update vector is disclosed to the server, aside from its aggregate. At the same time, communication efficiency should be achieved by leveraging the sparsity of the update vector, ensuring both privacy and reduced transmission overhead.

In this paper, we propose doubly efficient privacy-perserving recommender systems (GREC) that collectively address the above challenges. To reduce communication costs during the training stage, we design a secure aggregation protocol based on functional secret sharing Boyle et al. (2015; 2016), achieving succinct communication cost. Unlike methods based on dimension reduction, top-k sparsification, or quantization, our protocol reduces communication costs without compromising accuracy and privacy. For privacy-preserving inference, we introduce a user-side post-processing local differential privacy (LDP) algorithm, which performs cloud inference on users' privatized embeddings and subsequently allows users to locally post-processing the server response using their knowledge of raw inputs and specific noise. This approach effectively resolves the inherent trade-off between privacy and utility for traditional LDP approaches.

We defer the discussion of related work to appendix A.1. The contribution of our work can be summarized as follows:

(1) We develop a SecAgg protocol that achieves succinct communication by exploiting the sparsity of the embedding update matrix. Although existing SecAgg protocols incur communication overhead that scales linearly with model size, our approach significantly improves upon this by achieving a slower scaling rate as the model size grows.

(2) We design a cloud inference approach for recommender systems with LDP guarantee. This method significantly reduces users' computational and storage overheads relative to the full model inference required by existing FL protocols, and addresses the inherent trade-off between utility and privacy in traditional LDP-based methods.

## 2 BACKGROUND AND PRELIMINARIES

### 2.1 PROBLEM STATEMENT

In FedRec, a number of users want to jointly train a recommendation system based on their private data. Denote $\mathcal{U} = \{u_1, u_2, ..., u_n\}$ as the set of common users and $\mathcal{I} = \{i_1, i_2, ..., i_m\}$ as the set of items. Each user $u \in \mathcal{U}$ has a private interaction set $\mathcal{R}_u = \{(i, r_{u,i}) | i \in \mathcal{I}_u\} \subset [m] \times \mathbb{R}$, where $\mathcal{I}_u$ denotes the set of items rated by user $u$ and $r_{u,i}$ denotes the rating user $u$ gives to item $i$. Denote $X \in \mathbb{R}^{n \times l_x}$ and $Y \in \mathbb{R}^{m \times l_y}$, respectively, as the user and item feature matrix. Our goal is to generate a rating prediction that minimizes the squared deviation between actual and estimated ratings.

We focus on a class of RecSys that models low-dimensional latent factors for user and items Koren et al. (2009); Xue et al. (2017); Rendle (2010). The recommender fits a model $f$ comprising of $d$-dimensional latent factors (or embeddings) for user $P \in \mathbb{R}^{n \times d}$ and item $Q \in \mathbb{R}^{m \times d}$, along with the remaining parameters $\theta$. Denote $p_u \in \mathbb{R}^d$ and $q_i \in \mathbb{R}^d$ as the latent factors (or embeddings) for user $u$ and item $i$. A general form of the rating prediction can be expressed as:

$$\hat{r}_{u,i} = f(x_u, y_i; p_u, q_i, \theta) \tag{1}$$

, where $x_u \in \mathbb{R}^{l_x}$ and $y_i \in \mathbb{R}^{l_y}$ denote the feature vector for user $u$ and item $i$, and $\hat{r}_{u,i}$ is the estimated prediction for user $u$ on item $i$.

Denote $l(\cdot)$ as a general loss function. The model is trained by minimizing:

$$\mathcal{L} = \sum_{u,i} l(r_{u,i}, \hat{r}_{u,i}) \tag{2}$$

The remaining parameters $\theta$ typically include but are not limited to: (1) Feature extractors that convert user and item feature vectors into fixed size representations, denoted as $F_x : \mathbb{R}^{l_x} \to \mathbb{R}^{l_x \times d}$ and $F_y : \mathbb{R}^{l_y} \to \mathbb{R}^{l_y \times d}$, respectively. (2) The feed-forward layers within a deep neural network model.

In each training round, users locally update their private parameters $\Theta_p$ and upload their updates of public parameters $\mathbf{g}_{\Theta_s}$ to the server. To safeguard the privacy of individual gradients, the server employs SecAgg to aggregate the gradients from all active clients and update the public model $\Theta_s$.

### 2.2 FUNCTIONAL SECRET SHARING

Our protocol leverages functional secret sharing (FSS) Boyle et al. (2015; 2016) to optimize the communication payload. FSS secret shares a function $f : \{0, 1\}^n \to \mathbb{G}$, for some abelian group $\mathbb{G}$, into two functions $f_1, f_2$ such that: (1) $f(x) = \sum_{i=1}^{2} f_i(x)$ for any $x$, and (2) each description of $f_i$ hides $f$.

**Definition 2.1** (Function Secret Sharing). A function secret sharing (FSS) scheme with respect to a function class $\mathcal{F}$ is a pair of efficient algorithms (FSS.Gen, FSS.Eval):

- FSS.Gen($1^\lambda, f$): Based on the security parameter $1^\lambda$ and function description $f$, the key generation algorithm outputs a pair of keys, $(k_1, k_2)$.

- FSS.Eval($k_i, x$): Based on key $k_i$ and input $x \in \{0, 1\}^n$, the evaluation algorithm outputs party $i$'s share of $f(x)$, denoted as $f_i(x)$. $f_1(x)$ and $f_2(x)$ form additive shares of $f(x)$.

FSS scheme should satisfy the following informal properties (defined formally in Appendix A.3.2):

- **Correctness:** Given keys $(k_1, k_2)$ of a function $f \in \mathcal{F}$, it holds that FSS.Eval($k_1, x$) + FSS.Eval($k_2, x$) = $f(x)$ for any $x$.

- **Security:** Given keys $(k_1, k_2)$ of a function $f \in \mathcal{F}$, a computationally-bounded adversary that learns either $k_1$ or $k_2$ gains no information about the function $f$, except that $f \in \mathcal{F}$.

A naive form of FSS scheme is to additively secret share each entry in the truth-table of $f$. However, this approach results in each share containing $2^n$ elements. To obtain polynomial share size, nontrivial scheme of FSS has been developed for simple function classses, e.g., point functions Boyle et al. (2015; 2016). Our approach utilizes the advanced FSS scheme for the point function. In the following, we provide the formal definition of point function.

**Definition 2.2** (Point Function). For $\alpha \in \{0, 1\}^n$ and $\beta \in \mathbb{G}$, the point function $f_{\alpha,\beta} : \{0, 1\}^n \to \mathbb{G}$ is defined as $f_{\alpha,\beta}(\alpha) = \beta$ and $f_{\alpha,\beta}(x) = 0$ for $x \neq \alpha$.

## 2.3 DIFFERENTIAL PRIVACY

Differential privacy (DP) Dwork (2006); Cormode et al. (2018) is regarded as the gold standard for privacy protection. We introduce the definition of local differential privacy (LDP), a specific case of DP where the server is untrusted and data privatization is conducted by the client.

**Definition 2.3** (Local Differential Privacy). A randomized algorithm $\mathcal{M}$ is $(\epsilon, \delta)$-locally differentially private if it satisfies:

$$\Pr[\mathcal{M}(x) \in S] \leq e^{\epsilon} \Pr[\mathcal{M}(x') \in S] + \delta \tag{3}$$

for any inputs $x, x' \in D$ and any measurable subset subset $S \subseteq \text{Range}(\mathcal{M})$.

An essential property of DP algorithms is the post-processing immunity property, stating that the composition of any data-independent function with a $(\epsilon, \delta)$-DP mechanism will remain $(\epsilon, \delta)$-DP.

**Definition 2.4** (Post-Processing). Let $\mathcal{M} : \mathcal{D} \to \mathcal{Y}$ be an $(\epsilon, \delta)$-differentially private mechanism and $f : \mathcal{Y} \to \mathcal{Z}$ be any data-independent function. Then, $f \circ \mathcal{M}$ is $(\epsilon, \delta)$-differentially private.

It is important to note that the post-processing immunity only holds when $f$ is independent of the original data as well as the randomized mapping $\mathcal{M}$. From the utility perspective, this property implies a lower bound on the error rate of the results returned by any data-independent function $f$.

Recent works have developed post-processing mechanisms to enhance the accuracy of DP algorithms by leveraging prior knowledge of data. Wang et al. (2024) utilizes resampling techniques to improve the utility of DP synthetic data on downstream tasks. Balle & Wang (2018) post-processes the output of Gaussian mechanism with adaptive estimation technique. Split-and-Denoise Mai et al. (2024) divides the language model into a local encoder and a cloud encoder, and employs a user-side pre-trained model to denoise output embedding. To bypass the post-processing immunity, our work will use a user-side post-processing function that is dependent on the original data and mapping mechanism $\mathcal{M}$ without additional effort to collect prior knowledge.

## 3 METHODOLOGY

In this section, we provide a detailed description of our GREC, illustrated in Figure 1. In the training phase, our key design is a communication-efficient SecAgg algorithm to aggregate the gradients of item embedding of the model. In the inference phase, our key design is a computation and memory-efficient algorithm for privacy-preserving cloud-based recommendation. Our GREC ensures consistent privacy and efficiency throughout the training and inference stage.

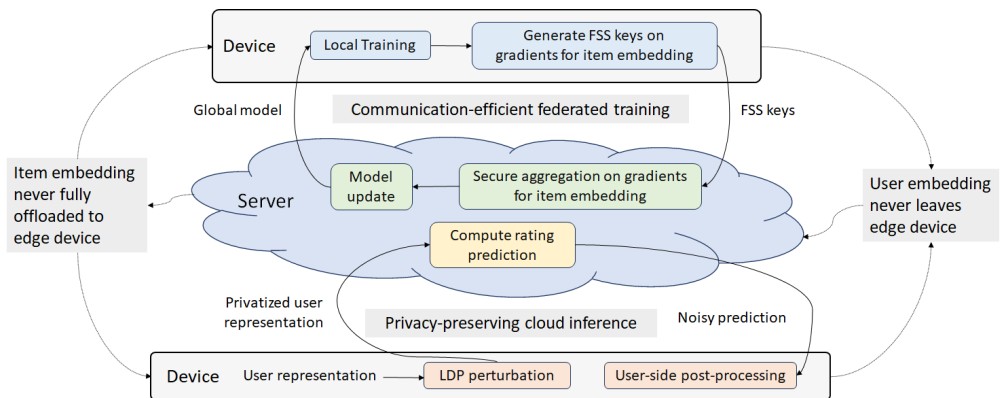

Figure 1: Overview of GREC, consisting of training and inference phase.

## 3.1 TRAINING

Observing that the public gradients primarily consist of highly sparse item embedding updates, we introduce a SecAgg protocol based on FSS to optimize the communication cost for the gradients of item embedding. Algorithm 2 outlines the training process for GREC.

### 3.1.1 KEY OBSERVATION

In most practical recommendation scenarios, the number of items a user has previously interacted with is typically much smaller than the total number of available items (see Figure 2(a)). This observation is linked to the information overload phenomenon, which recommender systems aim to address. Consequently, the gradient of the item embedding is zero for all items except those with which the user has previously interacted.

Denote $\mathbf{g}_Q$ as the gradient of item embedding $Q$, which is a sparse matrix. If using a general-purpose SecAgg, the communication cost to upload the sparse matrix is at least $O(bmd)$, where $b$ is number of bits required to represent a single numerical value. It is important to note that this corresponds to the bottleneck, since the embedding layer dominates the total model size as the item size increases (see Figure 2(b)).

Our goal is to optimize the communication cost for embedding layer, as this can significantly reduce the overall communication overhead, particularly under huge value of item size $m$.

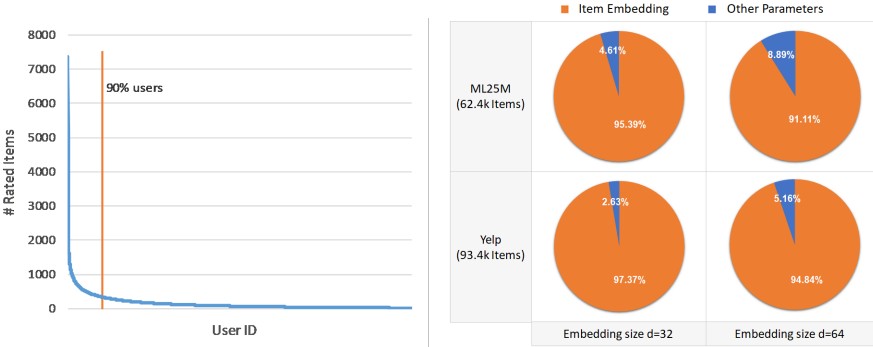

(a) Distribution of # rated items in ML10M      (b) Proportion of parameters in DeepFM

Figure 2: (a) The long-tailed distribution of the number of rated items in ML10M. The x-axis is the user id sorted by their activeness, and the y-axis represents the number of rated items for the user. For ML10M dataset with 27,278 items, nearly 90% users have rated only up to 300 items. (b) The proportion of item embeddings and other parameters within a three-layer deep factorization machine (DeepFM) for ML25M and Yelp, under two different embedding sizes $d = 32$ and $d = 64$.

### 3.1.2 SECURE AGGREGATION ON SPARSE UPDATE

Our key idea is to encode the sparse matrix (i.e., gradient of item embedding) $\mathbf{g}_Q \in \mathbb{R}^{m \times d}$ into some point functions. Then we can construct a 2-server secure aggregation (SecAgg) scheme based on the function secret sharing (FSS) of these point functions. We begin with the case where user $u$ rates a single item, i.e., $m'_u = 1$.

Suppose user $u$ has a sparse update $\mathbf{g}_{Q^u} \in \mathbb{R}^{m \times d}$ with only one non-zero row. Let $i$ denote the item index for the non-zero update. The SecAgg can be performed using the following steps:

**Step 1: Encode non-zero update with a point function, $\mathbf{g}_{Q^u} \to f_{u,i}$.** User $u$ begins by encoding $\mathbf{g}_{Q_i^u} \in \mathbb{R}^d$ with a point function $f_{u,i} : \mathcal{I} \to \mathbb{G}$, for some abelian group $\mathbb{G}$. The function $f_{u,i}$ takes an item id $x \in \mathcal{I}$ as input and outputs $f_{u,i}(x) = \mathbf{g}_{Q_i^u} \in \mathbb{R}^d$ if $x = i$, and $\mathbf{0} \in \mathbb{R}^d$ elsewhere.

**Step 2: Generate keys for the point function.** User $u$ secret shares the function $f_{u,i}$ with FSS scheme and outputs a pair of keys, i.e., $(k_u^1, k_u^2) = \text{FSS.Gen}(1^\lambda, f_{u,i})$. The keys $k_u^1$ and $k_u^2$ are sent to server 1 and 2, respectively.

**Step 3: Aggregate secret shares from users.** On receiving $k_u^s$ from all participating users, each server $s \in \{1, 2\}$ computes their secret shares of the aggregated matrix as follows:

$$\mathbf{v}_{Q_j}^s = \sum_u \text{FSS.Eval}(k_u^s, j), \ \ \forall j \in \mathcal{I} \tag{4}$$

**Step 4: Reconstruct gradient aggregation.** The two servers can collaborate to reconstruct the plaintext aggregation matrix. To be specific, server 2 sends the aggregated secret shares to server 1, and the plaintext aggregation can be recover by:

$$\mathbf{g}_Q = \mathbf{v}_Q^1 + \mathbf{v}_Q^2 \tag{5}$$

In the above procedure, user $u$ uploads only a single key to each server instead of the whole sparse vector. Additionally, the FSS security property ensures that each server learns no information about the rated item index $i$ and its gradient $\mathbf{g}_{Q_i^u}$. In the following, we extend the method to cases where $m_u' > 1$.

**SecAgg for $m_u' > 1$:** In step 1, user $u$ generates $m_u'$ point functions $f_{u,i} : \mathcal{I} \to \mathbb{G}$ for $i \in [m_u']$. Let $\text{idx}(x)$ denote the global index of the $i$-th rated item. Accordingly, $f_{u,i}$ takes a item id $x \in \mathcal{I}$ as input and outputs $f_{u,i}(x) = \mathbf{g}_{Q_{\text{idx}(i)}^u} \in \mathbb{R}^d$ if $x = i$, and $\mathbf{0} \in \mathbb{R}^d$ elsewhere. In step 2, user $u$ produces $m_u'$ pairs of secret keys $(k_{u,i}^1, k_{u,i}^2)$ for $i \in [m_u']$. In step 3, each server $s \in \{1, 2\}$ computes their secret shares of the aggregated matrix as follows:

$$\mathbf{v}_{Q_j}^s = \sum_u \sum_{i \in [m_u']} \text{FSS.Eval}(k_{u,i}^s, j), \ \ \forall j \in \mathcal{I} \tag{6}$$

The above design allows the user to transmit only $m_u'$ keys to each server without leaking the rated item index and their actual gradients. To further hide the number of rated items $m_u'$ from servers, we can pre-specify a unified update size $m'$ and accordingly pad or truncate the updated vectors to be an $m' \times d$ matrix (see Appendix A.4).

### 3.1.3 COMPLEXITY AND SECURITY ANALYSIS

**Complexity:** Denote $\lambda$ as the security parameter of FSS scheme, and $b$ as number of bits required to represent a single numerical value. The variables $d$ and $\theta$ refer to the embedding size and the parameters other than item embeddings. For a point function $f : \mathcal{X} \to \mathcal{Y}$, the user communication cost of function sharing $f$ is $|\mathcal{Y}| + (\lambda + 2) \log |\mathcal{X}|$, and the computation cost is $O(\log |\mathcal{X}| \cdot \text{AES})$, where AES denotes the complexity for each AES operation Boyle et al. (2016). In our specific case, the communication and computation costs for each point function are $(\lambda+2) \log m + bd$ and $O(\log m \cdot \text{AES})$, respectively. Considering $m'$ functions and $O(|\theta|)$ dense updates, we have communication complexity of $O(m'(bd + \lambda \log m) + |\theta|b)$, and computation cost of $O(m' \log m \cdot \text{AES} + |\theta|)$.

Table 1 compares the user-side cost between GREC and General-purpose SecAgg Xiong et al. (2020). We adopt the most efficient cost for General-purpose SecAgg (see Table 4). The communication cost of GREC scales linearly with $m'$ and logarithmically with $m$. GREC offers advantages over the General-purpose SecAgg scheme in terms of uplink communication cost as long as $m' < mbd / ((\lambda + 2) \log m + bd)$. The computation complexity of GREC primarily arises from AES operations, which can be mitigated when $m'$ is sufficiently small compared with $m$.

Table 1: User computation and communication cost of GREC and General-purpose SecAgg.

| | Communication Cost | Computation Cost |
|---|---|---|
| GREC | $O(m'(bd + \lambda \log m) + |\theta|b)$ | $O(m' \log m \cdot \text{AES} + |\theta|)$ |
| General-purpose SecAgg | $O(b(md + |\theta|))$ | $O(md + |\theta|)$ |

**Security:** The FSS security property ensures that the two non-colluding servers learn no more information than just the aggregated gradients. The FSS keys hide the rated item index as well as the values of updated gradients from each server. Under a pre-determined $m'$, servers are ignorant about the number of rated item for each user. Consequently, no additional information about the individual

updates is revealed to the servers. It is important to note that our algorithm can be integrated with DP to achieve stronger privacy protection. In particular, each server can independently add calibrated noises to the aggregated secret shares matrix, so that recovered aggregation matrix adheres to formal DP guarantee.

### 3.2 INFERENCE

#### 3.2.1 MOTIVATION

Existing protocols for FedRec neglect the computation cost and privacy concerns during inference, assuming that the user should maintain the full model for privacy-preserving inference, incurring a storage and memory cost of $O(md + \theta)$. Note that this cost is not inherent, even if we take into account the cost of training. During training, each user $u$ maintains only the rated item embeddings $\{Q_i\}_{i \in \mathcal{I}_u}$, their own user embedding $p_u$, and the remaining parameters $\theta$, resulting in $O(m'_u d + \theta)$ memory and storage cost. Therefore, considering the limited computational power of edge devices, a privacy-preserving cloud inference solution is highly desirable.

#### 3.2.2 A NAIVE LDP SOLUTION

We start with a naive LDP-based solution for cloud-based inference framework. In this approach, users perturb their user embeddings and user feature representations under the LDP guarantee and then transmit them to the server.

Before privatization, the user obtains a continuous representation of their hidden factors. In particular, user $u$ transforms the user feature $x_u$ into a $l_x \times d$ matrix $V_u$ using a local feature extractor $F_x$. This matrix is then concatenated with the user latent factor $p_u$ to form a $(l_x + 1) \times d$ representation matrix $H_u = [p_u; V_u]$. To satisfy LDP guarantee, the representation matrix is clipped to a maximum Frobenius norm of $B$ and noises drawn from normal distribution are added:

$$\bar{H}_u = H_u \cdot \min\{1, B/\|H_u\|_F\}; \ \tilde{H}_u = \bar{H}_u + Z_u \tag{7}$$

, where $\|\cdot\|_F$ denotes the Frobenius norm, and $Z_u \in \mathbb{R}^{(l_x+1) \times d}$ is a noise matrix with each element independently drawn from $\mathcal{N}(0, \sigma)$. Under $(\epsilon, \delta)$-LDP, $\sigma$ is set to $\sigma = B \cdot \sqrt{2 \cdot \log(1.25/\delta)}/\epsilon$.

The deviation between $H_u$ and $\tilde{H}_u$ is bounded by:

$$\mathbb{E}\left[\|H_u - \tilde{H}_u\|_F^2/|H_u|\right] \leq \mathbb{E}\left[\|H_u - \bar{H}_u\|_F^2/|H_u|\right] + \sigma^2$$
$$= \underbrace{\mathbb{E}\left[\|H_u - \bar{H}_u\|_F^2/|H_u|\right]}_{\text{clipping error}} + \underbrace{2B^2 \cdot \log(1.25/\delta)/\epsilon^2}_{\text{privatization error}} \tag{8}$$

Focusing on the privatization error, we find that the mean square error (MSE) increases polynomially with $B$ and inversely with $\epsilon$. Note that $B$ typically scales with the number of elements in $H_u$. The analysis indicates that the LDP mechanism could result in lower error bound when $l_x$ and $d$ are sufficiently small. An ideal case is $l_x = 0$, i.e., user $u$ merely transmits the privatized user embedding $p_u$ to the server. Then the user could make acceptable trade-off between prediction utility and privacy budget for low level of $d$.

#### 3.2.3 POST-PROCESSING LDP

The above LDP mechanism encounters a deteriorating performance as feature size $l_x$ and embedding size $d$ scale. Furthermore, the intrinsic trade-off between utility and privacy constrains users to opt for a reduced level of privacy budget $\epsilon$. The following proposition provides a lower bound of MSE for the rating prediction returned by the server Guo et al. (2022).

**Proposition 3.1.** *Let $\mathbf{r} \in \mathcal{R} \subseteq \mathbb{R}^m$ be the rating prediction obtained from non-privatized representation $H$, and let $\tilde{\mathbf{r}} \in \mathcal{R} \subseteq \mathbb{R}^m$ be the noisy prediction obtained under $(\epsilon, 0)$-privacy mechanism. Denote $F_s : \mathbb{R}^m \to \mathbb{R}^m$ as the server-side post-processing on the noisy rating prediction $\tilde{\mathbf{r}}$. Suppose $f_s$ is unbiased, then:*

$$\mathbb{E}[\|F_s(\tilde{\mathbf{r}}) - \mathbf{r}\|/m] \geq \frac{\sum_{i=1}^m \text{diam}_i(\mathcal{R})^2/4m}{e^\epsilon - 1} \tag{9}$$

*where $\text{diam}_i(\mathcal{R}) = \sup_{\mathbf{r}, \mathbf{r}' \in \mathcal{R}: r_j = r'_j \forall j \neq i} |r_i - r'_i|$ is the diameter of $\mathcal{R}$ in the $i$-th dimension.*

Proposition 3.1 reveals that the MSE lower bound increases inversely with $\epsilon$, regardless of any post-processing techniques the server applies to correct prediction errors. One critical limitation of server-side post-processing is that $F_s$ is independent of $H$ and $Z$, which constrains the server's capacity to conduct error correction on the results.

To overcome the fundamental barrier of post-processing immunity, we propose a user-side post-processing LDP approach, which leverages users' pre-stored noise matrices and non-privatized representations. Denote $F_p : (\mathcal{I}, \mathbb{R}^{2(l_x+1) \times d+1}) \to \mathbb{R}$ as a user-side post-processing function that takes the noisy rating $(i, \tilde{r}_{u,i})$, noise matrix $Z_u$, and clean representation matrix $H_u$ as inputs, and outputs a prediction $\hat{r}_{u,i}$ with lower expected error than $\tilde{r}_{u,i}$.

The post-processing function $F_p$ is parameterized by a lightweight denoise model. The denoise model is trained in an FL setting subsequent to the training of the recommender system, and deployed on user device in the inference phase. Since the the user representation is obtained during the training stage, there is no need to gather prior knowledge about the private input for denoise model training. Refer to Appendix A.8 for more details on the architecture and training of denoise model.

### 3.2.4 PRIVACY ANALYSIS

During inference, only the privatized representation matrix $\tilde{H}_u$ is transmitted to the server. Consequently, the server's view is limited to the privatized matrix $\tilde{H}_u$. In the following, we will demonstrate that the server's view adheres to the LDP guarantee. The process that privatizes the representation matrix $H_u$ into $\tilde{H}_u$, denoted as $M : \mathbb{R}^{(l_x+1) \times d} \to \mathbb{R}^{(l_x+1) \times d}$, satisfies LDP:

**Theorem 3.2.** *For any $\delta > 0$ and $\epsilon > 0$, the mechanism $M : \mathbb{R}^{(l_x+1) \times d} \to \mathbb{R}^{(l_x+1) \times d}$ achieves $(\epsilon, \delta)$-differential privacy.*

We proceed to analyze the security of the denoise model. During inference, the server is unable to infer the actual prediction using the denoise model. The inputs to the denoise model $F_p$ include the noise matrix and the clean embedding matrix, both of which are maintained privately on the user side. Consequently, the server does not have access to the critical inputs required for error correction.

## 4 EXPERIMENT EVALUATION

### 4.1 EXPERIMENT SETTING

We evaluate our GREC on five public datasets: MovieLens 100K (ML100K), MovieLens 1M (ML1M), MovieLens 10M (ML10M), MovieLens 25M (ML25M), and Yelp Harper & Konstan (2015); Yelp (2015). For the Yelp dataset, we sample a portion of top users ranked in descending order by their number of rated items, and obtain a subset containing 10,000 users and 93,386 items. Table 5 summarizes the statistics for the datasets.

Our framework is tested with four latent factor-based recommender models: matrix factorization with biased term (MF) Koren et al. (2009), neural collaborative filtering (NCF) He et al. (2017), factorization machine (FM) Rendle (2010), and deep factorization machine (DeepFM) Guo et al. (2017). Detailed hyperparameters for each model are provided in Appendix A.9.2.

### 4.2 COMMUNICATION ANALYSIS

To evaluate the communication efficiency of our framework, we conduct a comparative analysis of the communication payload during upload transmission between GREC and General-purpose SecAgg, as presented in Table 2. We use the two-server ASS, which has the minimal communication overhead, to compute the cost for General-purpose SecAgg (see Table 4). A key finding is that GREC's communication overhead increases at a significantly slower rate with item size compared to the two-server ASS, particularly for models characterized by a higher proportion of sparse updates.

Specifically, for MF and FM, which involve minimal dense updates, our protocol reduces the communication costs by approximately 4x to 90x, depending on the item size of the dataset. For NCF that includes a small share of dense updates, GREC achieves overhead reductions ranging from roughly 3.5x to 70x. For DeepFM, the reduction is less pronounced for the ML100K and ML1M datasets with item sizes lower than 4k, while the cost savings become more substantial as the item

size exceeds 10k. Note that our method is lossless, and the utility comparison with existing message compression methods is presented in the Appendix A.10.1.

Table 2: Communication cost (in MB) per user for GREC and General-purpose SecAgg during upload transmission in one iteration. Reduction ratio is given by the ratio of communication overhead of General-purpose SecAgg to that of GREC.

|  |  | ML100K (1.7k Items) | ML1M (3.9k Items) | ML10M (10.7k Items) | ML25M (62.4k Items) | Yelp (93.4k Items) |
|---|---|---|---|---|---|---|
| MF | General SecAgg | 0.87 | 2.02 | 5.55 | 32.46 | 48.56 |
|  | GREC | 0.17 | 0.27 | 0.28 | 0.51 | 0.52 |
|  | Reduction Ratio | 5.12 | 7.53 | 19.69 | 63.54 | 93.35 |
| NCF | General SecAgg | 0.45 | 1.03 | 4.20 | 24.48 | 30.64 |
|  | GREC | 0.13 | 0.20 | 0.26 | 0.46 | 0.43 |
|  | Reduction Ratio | 3.59 | 5.24 | 16.42 | 53.34 | 70.82 |
| FM | General SecAgg | 0.93 | 2.04 | 5.56 | 32.47 | 48.57 |
|  | GREC | 0.22 | 0.29 | 0.29 | 0.52 | 0.53 |
|  | Reduction Ratio | 4.13 | 6.98 | 19.03 | 62.29 | 91.90 |
| DeepFM | General SecAgg | 14.96 | 8.87 | 8.72 | 35.63 | 51.20 |
|  | GREC | 14.26 | 7.12 | 3.45 | 3.68 | 3.16 |
|  | Reduction Ratio | 1.05 | 1.25 | 2.53 | 9.69 | 16.20 |

## 4.3 INFERENCE UTILITY ANALYSIS

In Table 3, we examine the prediction accuracy of our GREC during inference phase in terms of RMSE under three settings: (1) *Non-private*, where raw features and user embeddings are directly transmitted to the server for cloud inference. (2) *Naive LDP*, where the representation matrix $H_u$ is privatized with LDP before sending to the server. (3) *Post-processing LDP*, the inference protocol proposed by our GREC.

Table 3: Inference accuracy in terms of RMSE. The privacy budget is fixed to $\epsilon = 1$ and $\delta = 10^{-4}$. Diff (%) is the percentage difference between naive LDP and post-processing LDP.

|  |  | ML100K | ML1M | ML10M | ML25M | Yelp |
|---|---|---|---|---|---|---|
| MF | Non-private | $0.944 \pm 0.003$ | $0.903 \pm 0.002$ | $0.868 \pm 0.003$ | $0.864 \pm 0.002$ | $1.050 \pm 0.001$ |
|  | Naive LDP | $1.693 \pm 0.021$ | $1.637 \pm 0.008$ | $1.403 \pm 0.009$ | $1.603 \pm 0.017$ | $1.580 \pm 0.008$ |
|  | **Post-processing LDP** | $0.957 \pm 0.007$ | $0.919 \pm 0.001$ | $0.875 \pm 0.001$ | $0.870 \pm 0.002$ | $1.097 \pm 0.001$ |
|  | Diff (%) | 43.47 | 43.86 | 37.63 | 45.73 | 30.57 |
| NCF | Non-private | $0.949 \pm 0.014$ | $0.897 \pm 0.006$ | $0.819 \pm 0.003$ | $0.786 \pm 0.008$ | $1.035 \pm 0.001$ |
|  | Naive LDP | $1.297 \pm 0.015$ | $1.169 \pm 0.068$ | $1.383 \pm 0.016$ | $1.496 \pm 0.020$ | $1.857 \pm 0.006$ |
|  | **Post-processing LDP** | $0.962 \pm 0.000$ | $0.915 \pm 0.005$ | $0.839 \pm 0.001$ | $0.812 \pm 0.000$ | $1.083 \pm 0.001$ |
|  | Diff (%) | 25.83 | 22.73 | 39.33 | 45.72 | 41.68 |
| FM | Non-private | $0.937 \pm 0.004$ | $0.906 \pm 0.000$ | $0.848 \pm 0.002$ | $0.789 \pm 0.003$ | $1.008 \pm 0.003$ |
|  | Naive LDP | $1.851 \pm 0.020$ | $2.350 \pm 0.018$ | $1.411 \pm 0.008$ | $2.042 \pm 0.009$ | $1.851 \pm 0.005$ |
|  | **Post-processing LDP** | $0.945 \pm 0.003$ | $0.908 \pm 0.000$ | $0.881 \pm 0.001$ | $0.813 \pm 0.003$ | $1.055 \pm 0.002$ |
|  | Diff (%) | 48.95 | 61.36 | 37.56 | 60.19 | 43.00 |
| DeepFM | Non-private | $0.939 \pm 0.006$ | $0.902 \pm 0.001$ | $0.821 \pm 0.001$ | $0.791 \pm 0.001$ | $1.011 \pm 0.002$ |
|  | Naive LDP | $2.120 \pm 0.018$ | $2.281 \pm 0.007$ | $1.703 \pm 0.011$ | $1.573 \pm 0.010$ | $1.776 \pm 0.008$ |
|  | **Post-processing LDP** | $0.943 \pm 0.001$ | $0.905 \pm 0.003$ | $0.833 \pm 0.002$ | $0.799 \pm 0.001$ | $1.055 \pm 0.002$ |
|  | Diff (%) | 55.52 | 60.32 | 51.09 | 49.21 | 40.60 |

It can be observed that our proposed GREC inference protocol significantly enhances the prediction performance via a user-side post-processing LDP, with little performance loss compared to the non-private setting. Specifically, employing the user-side post-processing function results in an average RMSE reduction of 40.3%, 34.9%, 50.2%, and 51.3% for MF, NCF, FM, and DeepFM, respectively. Compared with the non-private setting, the average decrease in accuracy remains within 2.2%.

## 4.4 COMPUTATION ANALYSIS

The computation time to generate the secret shares and FSS keys is presented in Figure 3. We utilize the computation cost for two-server ASS, which has the minimal computation overhead. GREC offers a computational advantage over the General-purpose SecAgg, as users need to generate shares only for the non-zero gradients of item embeddings rather than for the entire matrix.

In Figure 4, we compare the user-side computation time and memory cost between full model inference and post-processing LDP on four datasets. Using the post-processing LDP approach, the user's computation time and memory cost is reduced by, respectively, 11.48x and 7.32x, on average compared with the full model inference.

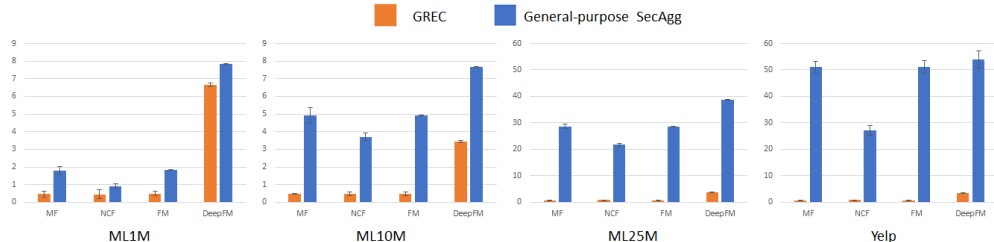

Figure 3: User computation time (in milliseconds) for secret shares generation during training phase.

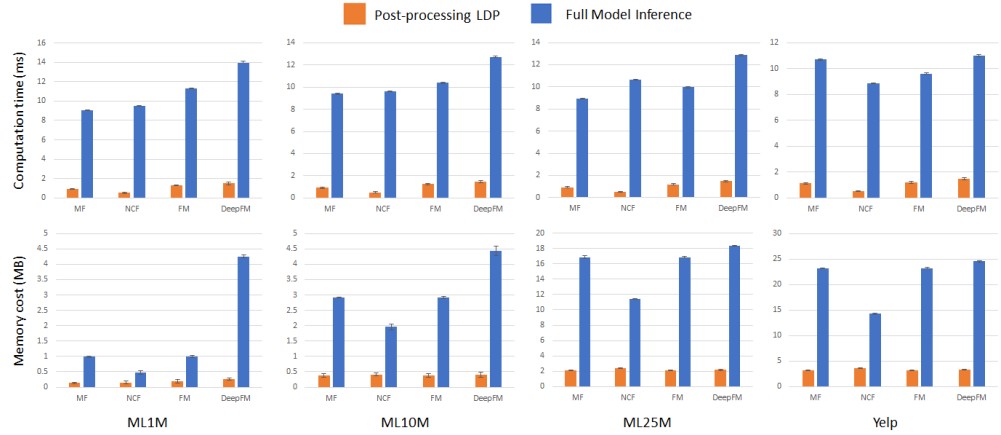

Figure 4: Computation time (in milliseconds) and memory cost (in MB) per user during inference phase. The computation cost is evaluated on the inference for 150 items. Full model inference deploys the entire model on user side for private inference.

## 5 CONCLUSION

This paper proposes a doubly efficient privacy-perserving recommender systems (GREC) to address the resource constraint of edge devices in terms of (a) upload bandwidth, and (b) computational power and storage. To reduce communication costs during upload transmission, we design a FSS-based SecAgg, achieving communication cost logarithmic in item size $m$. To reduce user computation burdens during inference, we introduce a post-processing LDP approach that addresses the intrinsic trade-off between privacy and utility. The empirical evaluation demonstrates that: (1) Our algorithm reduces communication costs by up to 90x compared with existing SecAgg protocols. (2) Our post-processing LDP approach enhances prediction accuracy by an average of 43.9% compared to standard LDP perturbation, while also reducing user-side computation time by approximately 11x relative to full model inference. Refer to Appendix A.11 for further discussions of our framework.

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

# A    APPENDIX

## A.1    RELATED WORK

### A.1.1    CROSS-USER FEDERATED RECOMMENDER SYSTEM

In recent years, federated recommender system (FedRec) trained on individual users has gained growing interest in research community. FCF Ammad-Ud-Din et al. (2019) and FedRec Lin et al. (2020) are among the pioneering implementations of federated learning for collaborative filtering based on matrix factorization. Privacy guarantees are enhanced through the application of cryptographic methods to the transmitted gradients Chai et al. (2020); Mai & Pang (2023). Difacto Li et al. (2016) introduces a distributed factorization machine algorithm that is scalable to a large number of users and items. FedNCF Perifanis & Efraimidis (2022) is a federated realization of neural collaborative iltering (NCF), where secure aggregation is leveraged to protect user gradients. FMSS Lin et al. (2022) proposes a federated recommendation framework for several recommendation algorithms based on factorization and deep learning. Rabbani et al. (2023) and Xu et al. (2022) improve the training efficiency for edge device using locality-sensitive hashing (LSH) techniques Chen et al. (2020; 2019). Despite the development of various algorithms for training FedRec systems, there remains a dearth of research investigating the inference phase.

### A.1.2    SECURE AGGREGATION FOR MACHINE LEARNING

Secure Aggregation (SecAgg) computes the summation of private gradients without revealing any individual values. Bonawitz et al. (2017) introduces a secure aggregation protocol for FL, leveraging a combination of pairwise masking, Shamir's Secret Sharing, and symmetric encryption techniques. Bell et al. (2020) reduces the communication and computation overhead to depend logarithmic in the client size. FastSecAgg Kadhe et al. (2020) designs a multi-secret sharing protocol based on Fast Fourier Transform to save computation cost. SAFELearn Fereidooni et al. (2021) designs an secure two-party computation protocol for efficient FL implementation. LightSecAgg So et al. (2022) reduces the computation complexity via one-shot reconstruction of aggregated mask. The two-server additive secret sharing (ASS) protocol Xiong et al. (2020) represents the most efficient SecAgg approach in terms of computation and communication complexity. Refer to Table 4 for the complexity of existing SecAgg algorithms. However, current SecAgg protocols incur communication costs that scale linearly with model size, and there is a lack of research on leveraging model update sparsity for enhanced efficiency.

## A.2    COMPLEXITY OF EXISTING SECAGG ALGORITHMS

In Table 4, we summarize the computation and communication complexity of existing SecAgg algorithms. It can be observed that a) two-server ASS represents the most efficient algorithm in term of both computation and communication complexity, and b) the per-client communication cost depends linear in the model size $l$ for all protocols.

Table 4: Computation and communication complexity of existing SecAgg algorithms. SecAgg and SecAgg+ refer to the algorithm proposed by Bonawitz et al. (2017) and Bell et al. (2020) respectively. $n$ and $l$ denote client size and model size, respectively.

| | Server | | Client | | Rounds |
|---|---|---|---|---|---|
| | Computation | Communication | Computation | Communication | |
| SecAgg | $O(n^2 l)$ | $O(nl + n^2)$ | $O(nl + n^2)$ | $O(l + n)$ | 4 |
| SecAgg+ | $O(nl \log n + n \log^2 n)$ | $O(nl + n \log n)$ | $O(l \log n + \log^2 n)$ | $O(l + \log n)$ | 3 |
| FastSecAgg | $O(l \log n)$ | $O(nl + n^2)$ | $O(l \log n)$ | $O(l + n)$ | 3 |
| LightSecAgg | $O(nl \log^2 n)$ | $O(nl)$ | $O(nl \log^2 n)$ | $O(nl)$ | 2 |
| SAFELearn | $O(nl)$ | $O(nl)$ | $O(l)$ | $O(l)$ | 2 |
| Two-server ASS | $O(nl)$ | $O(nl)$ | $O(l)$ | $O(l)$ | 1 |

### A.3 PRELIMINARIES

#### A.3.1 ADDITIVE SECRET SHARING

Additive secret sharing (ASS) Cramer et al. (2015) divides a secret $x \in \mathbb{F}_p$ from a finite field into $n$ shares, such that $\sum_{i=1}^n x_i \pmod{p} = x$. Consequently, any $n-1$ shares reveal nothing about the secret $s$. Furthermore, given two secret shares $[\![x]\!] = (x_1, ..., x_n)$ and $[\![y]\!] = (y_1, ..., y_n)$ from $\mathbb{F}_p$, it holds that $[\![x+y]\!] = (x_1 + y_1, ..., x_n + y_n)$.

#### A.3.2 FUNCTION SECRET SHARING

In this section we formally define the correctness and security properties of FSS scheme.

**Definition A.1** (FSS Correctness and Security). Let FSS = (FSS.Gen, FSS.Eval) be a FSS scheme for a function class $\mathcal{F}$, satisfying the following properties:

- **Correctness:** For every $x$ in the domain of $f$, it holds that:

$$\Pr\left(\sum_{i=1}^2 \text{FSS.Eval}(k_i, x) = f(x) \in \mathbb{F} : (k_1, k_2) \leftarrow \text{FSS.Gen}(1^\lambda, f)\right) = 1 \quad (10)$$

- **Security:** For any party $s \in \{1, 2\}$, there exists a PPT algorithm Sim (simulator), such that for every function $f \in \mathcal{F}$, the outputs of the following experiments REAL and IDEAL are computationally indistinguishable:

  - $\text{REAL}(1^\lambda, f) = \{k_s : (k_1, k_2) \leftarrow \text{FSS.Gen}(1^\lambda, f)\}$
  - $\text{IDEAL}(1^\lambda, f, \mathcal{F}) = \{k_s \leftarrow \text{Sim}(1^\lambda, \mathcal{F})\}$

### A.4 STANDARDIZATION OF UPLOADED ITEM SIZE

To conceal $m'_u$ from the server, a uniform $m'$ can be applied to all users. An optimal $m'$ should be substantially smaller than $m$ to reduce communication overhead, yet not excessively small to encompass the rated items of a majority of users. To determine a suitable value of $m'$, the server can compute the average number of rated items from all users via a SecAgg protocol and select $m'$ as follows:

$$m' = \alpha \cdot \frac{1}{n} \cdot \sum_u m'_u \quad (11)$$

, where $\alpha$ is a pre-specified multiplier on the average. Note that the SecAgg operation on the number of rated items is cheap, incurring $O(1)$ communication and computation overheads per user.

Given the unified $m'$, each user can standardize their non-zero updates for item embedding to be a $m' \times d$ matrix according to Algorithm 1.

---

**Algorithm 1** PadOrTrunc

---

> **Input:** $m'$ and $\mathbf{g}_{Q_u} \in \mathbb{R}^{m'_u \times d}$.
> **Output:** $\mathbf{g}'_{Q_u} \in \mathbb{R}^{m' \times d}$.
> **if** $m'_u < m'$ **then**
>     Create padding matrix of zero elements $\mathbf{0} \in \mathbb{R}^{(m' - m'_u) \times d}$
>     Concatenate $\mathbf{g}_{Q_u}$ and $\mathbf{0}$ to form $\mathbf{g}'_{Q_u} \in \mathbb{R}^{m' \times d}$
> **else if** $m'_u > m'$ **then**
>     Randomly sample $m'$ rows from $\mathbf{g}_{Q_u}$ to form $\mathbf{g}'_{Q_u} \in \mathbb{R}^{m' \times d}$
> **else**
>     Let $\mathbf{g}'_{Q_u} = \mathbf{g}_{Q_u}$
> **end if**
> **return** $\mathbf{g}'_{Q_u}$

---

---

**Algorithm 2** Federated Training of GREC

---

**Server** $s \in \{1, 2\}$:
**Initialize** public parameters $\Theta_s$.
**for** $t \in [1, T]$ **do**
    Distribute public parameters to users $u \in \mathcal{A}_t$.
    Receive FSS keys $\{k_{u,i}^s\}_{i \in [m']}$ and secret shares for dense update $\mathbf{v}_{\theta_u}^s$ from users $u \in \mathcal{A}_t$
    Compute the secret shares of the aggregated sparse update via equation 6.
    Aggregate the secret shares of the dense update via equation 12.
    **if** $i = 1$ **then**
        Receive the aggregated secret shares $(\mathbf{v}_Q^2, \mathbf{v}_\theta^2)$ from server 2
        Recover the gradients for public parameters $\mathbf{g}_Q, \mathbf{g}_\theta$.
        Update public parameters $\Theta_s = (Q, \theta)$ with the gradients.
    **else**
        Send the aggregated secret shares $(\mathbf{v}_Q^2, \mathbf{v}_\theta^2)$ to server 1.
    **end if**
**end for**

**User** $u \in [1, P]$:
**for** $t \in [1, T]$ **do**
    **if** $u \in \mathcal{A}_t$ **then**
        On receiving public parameters from server 1, read $(\theta, \{Q_i\}_{i \in \mathcal{I}_u})$ and discard $\{Q_i\}_{i \notin \mathcal{I}_u}$.
        Calculate gradients locally and update private parameters $\Theta_p$.
        Construct additive secret shares $(\mathbf{v}_{\theta_u}^1, \mathbf{v}_{\theta_u}^2)$ for dense gradient $\mathbf{g}_{\theta_u}$.
        Pad or truncate the sparse gradient into a $m' \times k$ matrix, $\mathbf{g}_{Q^u}' = \text{PadOrTrunc}(m', \mathbf{g}_{Q^u})$.
        Encode the sparse gradients with a point function, obtaining $\{f_{u,i}\}_{i \in [m']}$.
        Generate FSS keys for the sparse gradients $(k_{u,i}^1, k_{u,i}^2) = \text{FSS.Gen}(1^\lambda, f_{u,i})$ for $i \in [m']$.
        Send $(\mathbf{v}_{\theta_u}^i, \{k_{u,i}^s\}_{i \in [m']})$ to server $i \in \{1, 2\}$.
    **end if**
**end for**

---

## A.5 SECURE AGGREGATION ON DENSE UPDATE

We employ additive secret sharing for SecAgg on the dense update $\mathbf{g}_\theta$. In particular, user $u$ generates a pair of additive secret shares for the gradients $[\![\mathbf{g}_\theta]\!] = (\mathbf{v}_\theta^1, \mathbf{v}_\theta^2)$, and send the secret shares to the corresponding servers. Each server $s$ aggregates the secret shares from all participating users:

$$\mathbf{v}_\theta^s = \sum_u \mathbf{v}_{\theta_u}^s \tag{12}$$

Same as step 4 in Section 3.1.2, the two servers can subsequently collaborate to reconstruct the plaintext aggregated update.

## A.6 ALGORITHM TO TRAIN GREC

Algorithm 2 outlines the process to train the FedRec in a communication-efficient way.

## A.7 THEORETICAL EFFICIENCY ANALYSIS

**Communication cost**: In each iteration, the user uploads $m'$ FSS keys and secret shares of dense updates to the server. Each key size is $(\lambda + 2) \log m + bd$, where $b$ is number of bits required to represent a single numerical value Boyle et al. (2016). Thus total message size for FSS keys is $O(m'(bd + \lambda \log m))$. Furthermore, the size of additive shares for the dense updates is $O(|\theta|)$. Therefore, the total communication cost adds up to $O\left(m'(bd + \lambda \log m) + |\theta|b\right)$.

**Computation cost**: Each user generate $m'$ FSS keys and secret shares of dense updates. It takes $O(\log m \cdot \text{AES})$ operations to produce a FSS key Boyle et al. (2016), resulting complexity of $O(m' \log m \cdot \text{AES})$ for $m'$ keys. Additionally, generating additive secret shares takes $O(|\theta|)$ operations. Therefore, the total computation cost adds up to $O\left(m' \log m \cdot \text{AES} + |\theta|\right)$.

A.8 Post-processing LDP Framework

A.8.1 Framework Description

In our inference framework, users perturb their user embeddings and user feature representations under the LDP guarantee. Subsequently, the server performs cloud-based inference on the privatized user features and transmits the noisy predictions back to the users. Each user then applies local post-processing through a lightweight denoising model. Following we explain each components in detail.

**Feature extraction and perturbation:** User $u$ transforms the user feature $x_u$ into a $l_x \times d$ matrix $V_u$ using a local feature extractor $F_x$. This matrix is then concatenated with user latent factor $p_u$ to form a $(l_x + 1) \times d$ representation matrix $H_u = [p_u; V_u]$. The representation matrix is clipped to a maximum Frobenius norm of $B$ and noises drawn from normal distribution are added:

$$\bar{H}_u = H_u \cdot \min\left\{1, \frac{B}{\|H_u\|_F}\right\}; \ \tilde{H}'_u = \bar{H}_u + Z'_u \tag{13}$$

, where $\|\cdot\|_F$ denotes the Frobenius norm, and $Z'_u \in \mathbb{R}^{(l_x+1)\times d}$ is a noise matrix with each element independently drawn from $\mathcal{N}(0, \sigma)$. Under $(\epsilon, \delta)$-LDP, $\sigma$ is set as:

$$\sigma = \frac{B \cdot \sqrt{2 \cdot \log(1.25/\delta)}}{\epsilon} \tag{14}$$

To improve the performance, we clip the norm of $\tilde{H}'_u$ and store the calibrated noise matrix $Z_u$:

$$\tilde{H}_u = \tilde{H}'_u \cdot \min\left\{1, \frac{B}{\|\tilde{H}'_u\|_F}\right\}; \ Z_u = \tilde{H}'_u - H_u \tag{15}$$

**Server inference:** On receiving $\tilde{H}'_u$ from user $u$, the server computes the user's predicted preferences on items $\mathcal{I}_f \subseteq \mathcal{I}$, returning a set of noisy prediction $\{i, \tilde{r}_{u,i}\}_{i \in \mathcal{I}_f}$ to user $u$.

**Local denoise with user input:** User hosts a lightweight denoise model for error correction on the noisy prediction. Given the noisy rating $(i, \tilde{r}_{u,i})$, noise matrix $Z_u$, and clean representation matrix $H_u$, the denoise model output a prediction $\hat{r}_{u,i}$ with lower expected error. Mathematically, the process can be formulated as:

$$\hat{r}_{u,i} = F_p(i, \tilde{r}_{u,i}, H_u, Z_u) \tag{16}$$

, where $F_p : (\mathcal{I}, \mathbb{R}^{2(l_x+1)\times d+1}) \to \mathbb{R}$ denotes a user-side denoise model.

A.8.2 Training and Design of Denoise Model

The denoise model consists of four modules:

- $F_i^p : \mathcal{I} \to \mathbb{R}^{\hat{d}}$ that maps the item id $i$ to a $\hat{d}$-dimensional embedding vector $e_i$ for item-specific error correction.
- $F_h^p : \mathbb{R}^{(l_x+1)\times d} \to \mathbb{R}^{(l_x+1)\times \hat{d}}$ that transforms the clean representation matrix $H_u$ into a $\hat{d}$-dimensional matrix $T_{H_u}$.
- $F_z^p : \mathbb{R}^{(l_x+1)\times d} \to \mathbb{R}^{(l_x+1)\times \hat{d}}$ that transforms the noise matrix $Z_u$ into a $\hat{d}$-dimensional matrix $T_{Z_u}$.
- $F_o^p : \mathbb{R}^{(2l_x+3)\times \hat{d}} \to \mathbb{R}$ that takes the embedding vector $e_i$, transformed representation matrix $T_{H_u}$, and transformed noise matrix $T_{Z_u}$ as input, and outputs the corrected prediction $\hat{r}_{u,i}$.

We find that selecting a denoising dimension $\hat{d}$ much lower than $d$ suffices to attain accuracy levels comparable to those of non-private settings, thereby enabling the implementation of a more lightweight denoising model.

The denoise model is trained in an FL setting subsequent to the training of the recommender system. The lead server 1 maintains the global denoise model, and users compute the local updates using

their own data. In each iteration $t$, the user $u$ samples the noise matrix $Z_u^{(t)}$ and generates the corresponding noisy prediction $\{i, \tilde{r}_{u,i}^{(t)}\}_{i \in \mathcal{I}_u}$. The model is then updated according to the following objective function:

$$\mathcal{L} = \sum_{i \in \mathcal{I}_u} l \left( r_{u,i}, F_p^{(t)}(i, \tilde{r}_{u,i}^{(t)}, H_u^{(t)}, Z_u'^{(t)}) \right) \tag{17}$$

, where $F_p^{(t)}$ represents the global denoise model at iteration $t$.

We employ the same SecAgg protocol as that to train the recommender model to aggregate user gradients in a privacy-preserving manner. The SecAgg algorithm ensures that the server learns no more than the aggregated gradients. Additionally, given that the update vector size and the number of training epochs for the denoise model are significantly smaller than those of the recommender model, it is safe to expect that the denoise model leaks less user information than the recommender model.

### A.8.3 PROOF OF THEOREM 3.2

*Proof.* The process of adding noises from $\mathcal{N}\left(0, \frac{B \cdot \sqrt{2 \cdot \log(1.25/\delta)}}{\epsilon}\right)$ preserves $(\epsilon, \delta)$-LDP Dwork et al. (2014). The subsequent norm clipping of $\tilde{H}_u'$ preserves $(\epsilon, \delta)$-LDP based on post-processing property. Thus, the mechanism $M : \mathbb{R}^{(l_x+1) \times d} \to \mathbb{R}^{(l_x+1) \times d}$ satisfies $(\epsilon, \delta)$-LDP. □

### A.9 SPECIFICATIONS ON EXPERIMENTAL SETTING

All experiments are tested on a server with 4 NVIDIA L40 GPU (CUDA version 11.8). Below we detail the dataset pre-processing and hyperparameter in our evaluation.

### A.9.1 DATASET AND PRE-PROCESSING

For each dataset, we encode the user and item features into binary vectors for model training. The features we select for binary encoding are given as follows:

- ML100K: movie genre, user gender, user age, and user occupation.
- ML1M: movie genre, user gender, user age, and user occupation.
- ML10M: movie genre.
- ML25M: movie genre.
- Yelp: restaurant state.

The statistics of the datasets are listed in Table 5.

Table 5: Statistics of the datasets. Yelp refers to the subset sampled from the whole dataset.

|  | # Users | # Items | # Ratings | # User Features | # Item Features | Density |
|---|---|---|---|---|---|---|
| ML100K | 943 | 1,682 | 100,000 | 84 | 19 | 6.30% |
| ML1M | 6,040 | 3,883 | 1,000,209 | 30 | 18 | 4.26% |
| ML10M | 69,878 | 10,681 | 10,000,054 | 0 | 20 | 1.34% |
| ML25M | 162,541 | 62,423 | 25,000,095 | 0 | 20 | 0.25% |
| Yelp | 10,000 | 93,386 | 1,007,956 | 0 | 16 | 0.11% |

### A.9.2 HYPERPARAMETERS OF RECOMMENDER AND DENOISE MODEL

Each dataset is divided into 80% training and 20% testing data. For all cases, the recommender system is trained for 200 epochs, and the corresponding denoise model is trained for 50 epochs. Each user represents an individual client and 100 clients are selected in each iteration. The parameters are updated using Adaptive Moment Estimation (Adam) Kingma (2014) method. We use the combination of MSE and the regularization term as the loss function. The security parameter is set to $\lambda = 128$.

Table 6: Hyperparameters for federated training of recommender system.

|   |   | ML100K | ML1M | ML10M | ML25M | Yelp |
|---|---|---|---|---|---|---|
| MF | Embedding size | 64 | 64 | 64 | 64 | 64 |
|  | Learning rate | 0.025 | 0.025 | 0.01 | 0.01 | 0.01 |
|  | Regularization weight | 0.01 | 0.001 | 0.001 | 0.001 | 0.01 |
| NCF | Embedding size | 16 | 16 | 24 | 24 | 20 |
|  | Learning rate | 0.001 | 0.0001 | 0.001 | 0.001 | 0.001 |
|  | Regularization weight | 0.001 | 0 | 0 | 0 | 0 |
| FM | Embedding size | 64 | 64 | 64 | 64 | 64 |
|  | Learning rate | 0.025 | 0.025 | 0.005 | 0.005 | 0.01 |
|  | Regularization weight | 0.1 | 0.001 | 0 | 0 | 0.01 |
| DeepFM | Embedding size | 64 | 64 | 64 | 64 | 64 |
|  | Learning rate | 0.025 | 0.025 | 0.005 | 0.005 | 0.01 |
|  | Regularization weight | 0.1 | 0.001 | 0 | 0 | 0.001 |

Each experiment is run for four rounds and the average values are reported. Table 6 lists the specific hyperparameters for each dataset and model.

For NCF, we fix the the architecture of the neural network layers to $2d \rightarrow d \rightarrow d/2$. For DeepFM, the neural network layers are fixed to $(l_x + l_y + 2)d \rightarrow 4d \rightarrow 2d$. We set the number of selected items $m'$ for ML100K, ML1M, ML10M, ML25M, and yelp as 200, 300, 300, 500, and 500, respectively. Table 7 presents the size of sparse and dense parameters, corresponding to the item embedding (including item bias term) and the remaining parameters.

Table 7: Size of dense and sparse parameters. # Sparse, # Non-zero Spr., and # Dense denote, respectively, the size of sparse update, size of non-zero elements in sparse update, and size of dense update.

|   |   | ML100K (1.7k Items) | ML1M (3.9k Items) | ML10M (10.7k Items) | ML25M (62.4k Items) | Yelp (93.4k Items) |
|---|---|---|---|---|---|---|
| MF | # Sparse | 109,330 | 252,395 | 694,265 | 4,057,495 | 6,070,090 |
|  | # Non-zero Spr. | 6,893 | 10,764 | 9,295 | 9,945 | 6,552 |
|  | # Dense | 0 | 0 | 0 | 0 | 0 |
| NCF | # Sparse | 55,506 | 128,139 | 523,369 | 3,058,727 | 3,828,826 |
|  | # Non-zero Spr. | 3,499 | 5,465 | 7,007 | 7,497 | 4,133 |
|  | # Dense | 688 | 688 | 1,512 | 1,512 | 1,060 |
| FM | # Sparse | 109,330 | 252,395 | 694,256 | 4,057,495 | 6,070,090 |
|  | # Non-zero Spr. | 6,893 | 10,764 | 9,295 | 9,945 | 6,552 |
|  | # Dense | 6,696 | 3,121 | 1,301 | 1,301 | 1,041 |
| DeepFM | # Sparse | 109,330 | 252,395 | 694,256 | 4,057,495 | 6,070,090 |
|  | # Non-zero Spr. | 6,893 | 10,764 | 9,295 | 9,945 | 6,552 |
|  | # Dense | 1,761,065 | 856,370 | 395,798 | 395,798 | 330,002 |

The hyperparameter to train the denoise model specific to each model and dataset is presented in Table 8.

## A.10    ADDITIONAL EXPERIMENT EVALUATION

### A.10.1    COMPARISON WITH EXISTING MESSAGE COMPRESSION METHODS

Given that existing message compression methods degrade the model performance, we evaluate the utility of GREC against several baselines on the federated training of MF, including: (1) Federated Matrix Factorization with SVD (FedMF w/ SVD) Nguyen et al. (2024), (2) Correlated Low-rank Structure (CoLR) Nguyen et al. (2024), (3) Federated Matrix Factorization with Top-K Sparsification (FedMF w/ Top-K) Gupta et al. (2021), and (4) Ternary Quantization (TernQuant) Wen et al. (2017).

Table 8: Hyperparameters for federated training of denoise model.

|  |  | ML100K | ML1M | ML10M | ML25M | Yelp |
|---|---|---|---|---|---|---|
| MF | Denoise dimension $\hat{d}$ | 8 | 8 | 8 | 8 | 8 |
|  | Learning rate | 0.025 | 0.01 | 0.01 | 0.01 | 0.01 |
|  | Regularization weight | 0.001 | 0.001 | 0.001 | 0.0001 | 0.001 |
| NCF | Denoise dimension $\hat{d}$ | 5 | 5 | 6 | 6 | 5 |
|  | Learning rate | 0.025 | 0.01 | 0.01 | 0.01 | 0.01 |
|  | Regularization weight | 0.001 | 0.001 | 0.001 | 0.0001 | 0.001 |
| FM | Denoise dimension $\hat{d}$ | 8 | 8 | 8 | 8 | 8 |
|  | Learning rate | 0.025 | 0.01 | 0.01 | 0.01 | 0.01 |
|  | Regularization weight | 0.01 | 0.001 | 0.001 | 0.0001 | 0.001 |
| DeepFM | Denoise dimension $\hat{d}$ | 8 | 8 | 8 | 8 | 8 |
|  | Learning rate | 0.025 | 0.01 | 0.01 | 0.01 | 0.01 |
|  | Regularization weight | 0.001 | 0.001 | 0.001 | 0.0001 | 0.0001 |

The first two methods represent dimension reduction approaches, the third employs the Top-K sparsification technique, and the fourth employs gradient quantization method.

Table 9 presents the prediction accuracy on ML10M and Yelp dataset. The embedding size is set to 64 for all cases. Noted that for consistency with the baselines, the bias term is not included in the MF model, leading to slightly different RMSE and Reduction ratio for GREC. It can be observed that: (1) FedMF w/ SVD and CoLR's abilities to reduce the communication cost is limited by the embedding size, while TernQuant's reduction ratio is limited by the default 32-bit precision. GREC has an advantage on reducing the communication cost by a large factor under higher value of item size. (2) Under similar reduction ratio, the performance is degraded on an average by 7.2%, 16.3%, 13.7%, and 29.9% for FedMF w/ SVD, CoLR, FedMF w/ Top-K, and TernQuant, respectively.

Table 9: RMSE and reduction ratio for various message compression methods on ML10M and Yelp. Reduction ratio refers to the ratio of uplink communication cost before and after the application of the compression mechanism. The values for RMSE denote the mean $\pm$ standard deviation of four rounds of experiments.

|  |  | GREC | FedMF w/ SVD | CoLR | FedMF w/ Top-K | TernQuant |
|---|---|---|---|---|---|---|
| ML10M | RMSE | $0.894_{\pm 0.004}$ | $0.903_{\pm 0.002}$ | $0.931_{\pm 0.002}$ | $0.906_{\pm 0.003}$ | $1.631_{\pm 0.007}$ |
|  | Reduction Ratio | 19.25 | 16.00 | 16.00 | 16.00 | 16 |
| Yelp | RMSE | $1.353_{\pm 0.004}$ | $1.563_{\pm 0.007}$ | $1.894_{\pm 0.011}$ | $1.829_{\pm 0.005}$ | $1.587_{\pm 0.006}$ |
|  | Reduction Ratio | 91.18 | 16.00 | 16.00 | 16.00 | 16 |

### A.10.2 INFERENCE OVERHEAD ANALYSIS

In Figure 5, we compared the storage cost for a user during inference in two cases: (1) The user maintains the entire recommender model for local inference. (2) The post-processing LDP protocol in our proposed GREC framework. It can by observed that our post processing LDP reduces the storage cost by over 7x on average compared with full model inference.

### A.10.3 TRAINING MEMORY AND STORAGE ANALYSIS

In Figure 6 we present the training memory and storage cost for two cases: (1) GREC where the user utilizes merely the related item embeddings for model training. (2) Full model training where user maintain the full model for training. It can be observed that GREC leads to substantial saving in memory and storage cost when the sparse item embedding matrix dominates the model parameters. For memory cost, the average savings are 12x, 21x, 101x, and 214x for ML1M, ML10M, ML25M, and Yelp, respectively. For storage cost, the average savings are 13x, 23x, 111x, and 247x for ML1M, ML10M, ML25M, and Yelp, respectively.

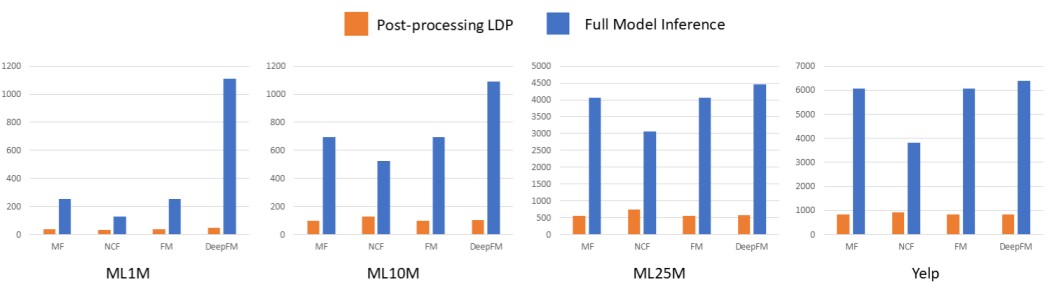

Figure 5: Storage cost (in $10^3$ parameters) per user during inference phase.

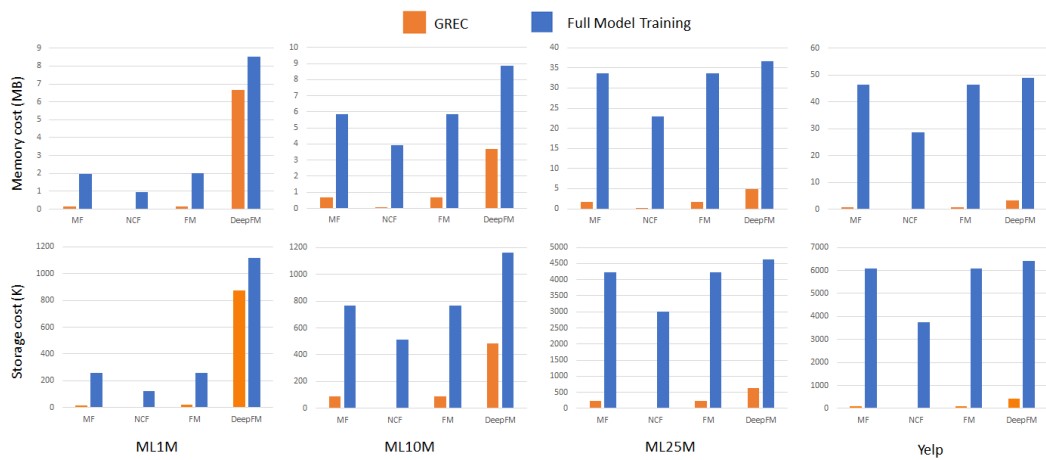

Figure 6: Average memory cost (in MB) and storage cost (in $10^3$ parameters) per user during training phase. The memory cost is computed with batch size of 1.

### A.10.4 UTILITY ANALYSIS UNDER HIGH DIMENSIONAL SETTING

To evaluate the robustness and scalability of our user-side post-processing method, we examine the impact of embedding dimension $d$ on the prediction accuracy. We utilize the same hyperparameters in Table 8 to train the denoise model, except that the denoise dimension $\hat{d}$ follows the specification in Table 10.

Table 10: Hyperparameter for denoise model training under various dimensions.

|  |  | | ML1M | | | Yelp | |
| --- | --- | --- | --- | --- | --- | --- | --- |
| MF | Embedding dimension $d$ | 64 | 128 | 512 | 64 | 128 | 512 |
|  | Denoise dimension $\hat{d}$ | 8 | 10 | 12 | 8 | 10 | 12 |
| NCF | Embedding dimension $d$ | 16 | 64 | 128 | 20 | 64 | 128 |
|  | Denoise dimension $\hat{d}$ | 5 | 6 | 8 | 5 | 6 | 8 |
| FM | Embedding dimension $d$ | 64 | 128 | 512 | 64 | 128 | 512 |
|  | Denoise dimension $\hat{d}$ | 8 | 10 | 12 | 8 | 10 | 12 |
| DeepFM | Embedding dimension $d$ | 64 | 128 | 512 | 64 | 128 | 512 |
|  | Denoise dimension $\hat{d}$ | 8 | 10 | 12 | 8 | 10 | 12 |

Table 11 presents the inference accuracy under higher embedding dimensions. The results reveal that our post-processing LDP can effectively maintain the inference utility for dimension $d$ up to

512. In particular, our post-processing LDP outperforms naive LDP by an average of 44% and 47% , respectively, for $d = 128$ and $d = 512$.

Table 11: Inference accuracy in terms of RMSE under various embedding dimensions. The privacy budget is fixed to $\epsilon = 1$ and $\delta = 10^{-4}$.

| | | ML1M | | | Yelp | | |
|---|---|---|---|---|---|---|---|
| | Dimension $d$ | 64 | 128 | 512 | 64 | 128 | 512 |
| MF | Non-private | 0.903 | 0.906 | 0.908 | 1.050 | 1.051 | 1.048 |
| | Naive LDP | 1.637 | 1.643 | 1.643 | 1.580 | 1.532 | 1.533 |
| | **Post-processing LDP** | 0.919 | 0.919 | 0.921 | 1.097 | 1.079 | 1.082 |
| NCF | Dimension $d$ | 16 | 64 | 128 | 20 | 64 | 128 |
| | Non-private | 0.897 | 0.896 | 0.909 | 1.035 | 1.034 | 1.039 |
| | Naive LDP | 1.169 | 1.241 | 1.380 | 1.857 | 1.563 | 1.636 |
| | **Post-processing LDP** | 0.915 | 0.922 | 0.921 | 1.083 | 1.059 | 1.097 |
| FM | Dimension $d$ | 64 | 128 | 512 | 64 | 128 | 512 |
| | Non-private | 0.906 | 0.908 | 0.905 | 1.008 | 1.010 | 1.006 |
| | Naive LDP | 2.350 | 2.313 | 2.335 | 1.851 | 1.859 | 1.856 |
| | **Post-processing LDP** | 0.908 | 0.908 | 0.906 | 1.055 | 1.056 | 1.056 |
| DeepFM | Dimension $d$ | 64 | 128 | 512 | 64 | 128 | 512 |
| | Non-private | 0.903 | 0.901 | 0.901 | 1.011 | 1.019 | 1.003 |
| | Naive LDP | 2.275 | 2.342 | 2.345 | 1.776 | 1.850 | 1.848 |
| | **Post-processing LDP** | 0.905 | 0.905 | 0.903 | 1.055 | 1.054 | 1.056 |

### A.10.5 UTILITY ANALYSIS UNDER VARIOUS PRIVACY BUDGETS

In this section, we study the impact of privacy budget $\epsilon$ on the inference utility. We vary the privacy budget from 0.1 to 10 in Table 12. Though the accuracy for naive LDP degrades significantly as $\epsilon$ decreases to 0.1, the performance for our post-processing LDP remains robust, with average utility loss of 2.9% compared to non-private setting for $\epsilon = 0.1$.

Table 12: Inference accuracy in terms of RMSE under various privacy budget $\epsilon$.

| | | ML1M | | | Yelp | | |
|---|---|---|---|---|---|---|---|
| | Privacy budget $\epsilon$ | 0.1 | 1 | 10 | 0.1 | 1 | 10 |
| MF | Non-private | | 0.903 | | | 1.050 | |
| | Naive LDP | 1.698 | 1.637 | 1.105 | 1.632 | 1.580 | 1.109 |
| | **Post-processing LDP** | 0.922 | 0.919 | 0.917 | 1.097 | 1.097 | 1.090 |
| NCF | Non-private | | 0.897 | | | 1.035 | |
| | Naive LDP | 1.164 | 1.169 | 1.138 | 1.862 | 1.857 | 1.870 |
| | **Post-processing LDP** | 0.913 | 0.915 | 0.911 | 1.084 | 1.083 | 1.075 |
| FM | Non-private | | 0.906 | | | 1.008 | |
| | Naive LDP | 2.377 | 2.350 | 2.075 | 2.001 | 1.851 | 1.076 |
| | **Post-processing LDP** | 0.908 | 0.908 | 0.905 | 1.059 | 1.055 | 1.036 |
| DeepFM | Non-private | | 0.903 | | | 1.011 | |
| | Naive LDP | 2.306 | 2.275 | 1.979 | 1.832 | 1.776 | 1.321 |
| | **Post-processing LDP** | 0.911 | 0.907 | 0.904 | 1.054 | 1.055 | 1.048 |

### A.10.6 COMPARISON WITH SPARSE AGGREGATION PROTOCOL

In this section, we discuss the advantages of our GREC over existing sparse aggregation protocols. We consider two SOTA frameworks, Secure Aggregation with Mask Sparsification (SecAggMask) Liu et al. (2023) and Top-k Sparse Secure Aggregation (TopkSecAgg) Lu et al. (2023). The key

problem with the two frameworks is that they fail to ensure that the server learns nothing except the aggregated gradients. In particular:

- Leakage of rated item index. For SecAggMask, each user transmits the union of gradients with non-zero updates and masks to the server. For TopkSecAgg, each user is required to upload the coordinate set of non-zero gradients along with a small portion of perturbed coordinates. In both methods, the server could narrow down the potential rated items to a much smaller set.

- Leakage of gradient values. While TopkSecAgg protects the values of non-zero updates against the server, SecAggMask would reveal the plaintext values to the server. Specifically, SecAggMask randomly masks a portion of the gradients to reduce communication cost, and fails to ensure that all non-zero gradients would be masked against any attackers.

In Table 13 we compare the communication cost of our GREC with the sparse aggregation protocols under the same training setting. For SecAggMask, we adopt a mask threshold such that 60% non-zero gradients would be masked in expectation. For TopkSecAgg, we set the purturbation proportion $\mu$ to be 0.1, following Lu et al. (2023). Both approaches result in higher communication cost than GREC because: (1) Besides the non-zero embedding gradients, SecAggMask requires the user to send a certain proportion of randomly masked zero updates to the server. (2) To cancel out the mask values, in TopkSecAgg each user sends the union of rated item embeddings for all participating user, rather than the those for each single user.

Table 13: Communication cost (in MB) per user for GREC and Sparse SecAgg during upload transmission in one iteration.

| | | ML100K (1.7k Items) | ML1M (3.9k Items) | ML10M (10.7k Items) | ML25M (62.4k Items) | Yelp (93.4k Items) |
|---|---|---|---|---|---|---|
| MF | General SecAgg | 0.87 | 2.02 | 5.55 | 32.46 | 48.56 |
| | SecAggMask | 0.27 | 0.61 | 1.66 | 9.60 | 14.35 |
| | TopkSecAgg | 0.32 | 0.66 | 0.91 | 1.17 | 1.95 |
| | **GREC** | 0.17 | 0.27 | 0.28 | 0.51 | 0.52 |
| NCF | General SecAgg | 0.45 | 1.03 | 4.20 | 24.48 | 30.64 |
| | SecAggMask | 0.14 | 0.31 | 1.25 | 7.21 | 8.98 |
| | TopkSecAgg | 0.17 | 0.34 | 0.69 | 0.89 | 1.23 |
| | **GREC** | 0.13 | 0.20 | 0.26 | 0.46 | 0.43 |
| FM | General SecAgg | 0.93 | 2.04 | 5.56 | 32.47 | 48.57 |
| | SecAggMask | 0.27 | 0.61 | 1.66 | 9.60 | 14.35 |
| | TopkSecAgg | 0.32 | 0.66 | 0.91 | 1.17 | 1.95 |
| | **GREC** | 0.22 | 0.29 | 0.29 | 0.52 | 0.53 |
| DeepFM | General SecAgg | 14.96 | 8.87 | 8.72 | 35.63 | 51.20 |
| | SecAggMask | 14.30 | 7.44 | 4.81 | 12.76 | 16.99 |
| | TopkSecAgg | 14.36 | 7.49 | 4.07 | 4.32 | 4.58 |
| | **GREC** | 14.26 | 7.12 | 3.45 | 3.68 | 3.16 |

### A.10.7 PERFORMANCE FOR ITEM RECOMMENDATION TASK

Previous experiments focus on the performance in terms of rating prediction, i.e., how close the predicted rating is to the actual rating. Here we evaluate the model performance in terms of item recommendation ability. In particular, how many items are selected by the user in the recommendation list. We utilize Hit Ratio (HR) and Normalized Discounted Cumulative Gain (NDCG) He et al. (2015) to measure the performance. During evaluation, we follow Koren (2008) to randomly samples 100 unrated items, and rank the test items along with the samples.

Table 12 shows the recommendation performance in terms of HR@10 and NDCG@10. We can observe that our post-processing LDP outperforms naive LDP by an average of 43% and 46%, respectively, for HR@10 and NDCG@10. Compared with the non-private setting, the utility loss of our method is 1.08% and 2.5%, respectively, for HR@10 and NDCG@10.

Table 14: Recommendation accuracy in terms of HR@10 and NDCG@10 for ML1M dataset. The privacy budget is fixed to $\epsilon = 1$ and $\delta = 10^{-4}$. Diff (%) is the percentage difference between naive LDP and post-processing LDP.

|  |  | MF | NCF | FM | DeepFM |
|---|---|---|---|---|---|
| HR@10 | Non-private | 0.593 | 0.591 | 0.605 | 0.604 |
|  | Naive LDP | 0.345 | 0.389 | 0.372 | 0.241 |
|  | **Post-processing LDP** | 0.586 | 0.584 | 0.598 | 0.599 |
|  | Diff (%) | 41.13 | 33.39 | 37.79 | 59.77 |
| NDCG@10 | Non-private | 0.337 | 0.334 | 0.347 | 0.344 |
|  | Naive LDP | 0.182 | 0.207 | 0.209 | 0.121 |
|  | **Post-processing LDP** | 0.333 | 0.313 | 0.343 | 0.339 |
|  | Diff (%) | 45.35 | 33.87 | 39.07 | 64.31 |

### A.10.8 SERVER COMPUTATION COST

To validate the practicality of GREC, we evaluate the server computation cost under increasing number of devices during the training stage. Noted that the server computation cost during the inference stage is the same as that in the centralized setting. Table 15 compares the computation time of our framework with homomorphic encryption (HE) approaches with CKKS cryptosystem Cheon et al. (2017). Though both frameworks scales linearly with the number of participating devices, GREC is approximately 130x faster than the typical HE protocol on average.

Table 15: Server computation cost (in minutes) per iteration for ML1M dataset.

| # of Active Users | 100 | 200 | 300 | 400 | 500 |
|---|---|---|---|---|---|
| HE (CKKS) | 127.26 | 244.51 | 391.76 | 519.05 | 646.38 |
| **GREC** | 0.99 | 1.92 | 2.93 | 3.88 | 4.90 |

### A.10.9 BREAKPOINT ANALYSIS OF COMMUNICATION COST

In Section 4.2 we show that GREC offers advantages over the General-purpose SecAgg scheme as long as $m' < mbd/((\lambda + 2) \log m + bd)$. The inequality usually holds for recommender system with sparse update.

Table 16 presents the maximum number of $m'$ for each dataset where the aforementioned inequality holds. We use security parameter $\lambda = 128$ and 32-bit precision $b = 32$. It can be observed that the breakpoint of $m'$ is sufficiently large, over 50% of the total item size $m$. It is highly improbable for a user to rate such a substantial proportion of items in practical scenarios.

Table 16: Maximum Value of $m'$ for Communication Cost Advantage over General-purpose SecAgg under Various Embedding Dimension $d$.

|  | ML100K (1.7k Items) | ML1M (3.9k Items) | ML10M (10.7k Items) | ML25M (62.4k Items) | Yelp (93.4k Items) |
|---|---|---|---|---|---|
| $d = 64$ | 1001 | 2210 | 5775 | 31038 | 45418 |
| $d = 128$ | 1255 | 2817 | 6408 | 37453 | 56031 |
| $d = 512$ | 1550 | 3547 | 9655 | 55418 | 82495 |

## A.11 DISCUSSION

**Communication cost during download transmission:** Our framework focuses on the overhead optimization during upload transmission considering its limited bandwidth. During the download stage, succinct communication cost can be achieved using private information retrieval (PIR) Chor et al. (1998) techinques, where users can retrieve their related item embeddings without revealing the item index. Existing PIR protocols can achieve communication costs that depend sublinearly on $m$

Chor & Gilboa (1997); Cachin et al. (1999). The use of FSS schemes further enhances communication efficiency in PIR, reducing overheads to logarithmic dependence on $m$ Boyle et al. (2015); Gilboa & Ishai (2014).

**Private inference for sequential recommendation:** Sequential recommendations predict the next item the user is likely to interact with given their interaction histories. Various models have been proposed for this recommendation task, based on architectures including Recurrent Neural Network (RNN), self-attention blocks, and Graph Neural Network (GNN) Wang et al. (2020); Kang & McAuley (2018); Sun et al. (2019). It's essential to develop a unified federated training and private inference for sequential recommendation compatible with a variety of models.

