# OpenReview forum: "GREC: Doubly Efficient Privacy-preserving Recommender Systems for Resource-Constrained Devices"
_ICLR.cc/2025/Conference — Submitted to ICLR 2025_

### Official Review · Reviewer_cpnX · 2024-10-31

**Soundness:** 3
**Presentation:** 3
**Contribution:** 3
**Rating:** 6
**Confidence:** 2

**Summary:**

This work proposes a privacy-preserving federated recommendation system method that aims to alleviate communication and computational bottlenecks that arise on resource-constrained edge devices. Communication costs are reduced via a functional secret sharing method, computational costs are reduced by shifting inference to the cloud, and privacy is ensured through a local differential privacy (LDP) mechanism. Empirical results showcase a reduction in memory while also reducing test RMSE.

**Strengths:**

I enjoy the idea of leveraging the central server for both privacy benefits (via the FSS) and computational costs. More FL algorithms should think about leveraging the central server for computational costs instead of vice-versa.

This method seems to be applicable to many other ML/FL applications where there are sparse updates to embedding layers.

The empirical reduction in communication costs is impressive and the reduction in RMSE is only very marginal.

The paper, while pretty technical, is well-written with nice presentation.

**Weaknesses:**

There aren't any true test accuracy results, could the authors report classification accuracy?

Is the method scalable, since the server has to receive FSS updates, perform secure aggregation, update the model, and perform inference all at once? It seems that there would be quite a bottleneck once the number of devices increase to realistic numbers.

**Questions:**

It would be nice if the authors could include the Related Works section within the main body. It would be easier for people, like me, to better grasp the problem and see other works if it is available in the main body.

If server's collude, will FSS fail in terms of privacy? What would motivate two separate servers to participate in training and take on extra computational burdens? In most settings, one single company would act as the server. When would two separate servers be a realistic setting?

For clarification, is $m$ simply the number of items that can be recommended? I was curious when the inequality $m′ < mbd/ ((λ + 2) log m + bd)$ would fail to hold. Is this ever a concern?

I know that there are some works that leverage locality-sensitive hashing approaches (LSH) to efficiently perform large-scale recommender system training. These include SLIDE and MONGOOSE (Chen et al. 2020/2021). Other works have leveraged these methods for LSH RecSys training in the edge setting, namely "Adaptive Sparse Federated Learning in Large Output Spaces via Hashing" (Xu et al. 2022) and "Large-Scale Distributed Learning via Private On-Device Locality-Sensitive Hashing" (Rabbani et al. 2023). These works might not be crucial to compare against, but they are another method for efficient RecSys training! Hope these can be interesting reads for you.

---

> ### Author Response · Authors · 2024-11-21
> **Rebuttal (Part-I)**
>
> We thank the reviewer for the recognizing the value of our paper. Hope our response below could address the reviewer's concern.
>
> **W1: There aren't any true test accuracy results, could the authors report classification accuracy?**
>
> Thanks for your comment and suggestion. Our paper focus on the rating prediction task, and thus measures the RMSE between predicted and true rating. For the classification accuracy, we think you are referring to **the proportion of items interated by a user among a recommendation list**. To measure this, we added experiments using two metrics:
> - HR@10: Hit Ratio measures whether the test item is present on the top-10 list.
> - NDCG@10: Normalized Discounted Cumulative Gain considers the position of the hit by assigning higher scores to hits at top ranks.
>
> The test accuracies is given in Table R1. **We can observe that our post-processing LDP outperforms naive LDP by an average of 43\% and 46\%, respectively, for HR@10 and NDCG@10.**
>
> [**Table R1.**  Recommendation accuracy in terms of HR@10 and NDCG@10 for ML1M dataset. ]
> | | |MF | NCF | FM | DeepFM |
> |-|-|-|-|-|-|
> |HR@10|Non-private| 0.593| 0.591 | 0.605 | 0.604 |
> | |Naive LDP|0.345| 0.389 | 0.372 | 0.241|
> | |Post-processing LDP|0.586| 0.584 | 0.598 |0.599|
> |NDCG@10|Non-private| 0.337| 0.334 | 0.347 | 0.344 |
> | |Naive LDP|0.182| 0.207 |0.209 |0.121|
> | |Post-processing LDP|0.333| 0.313 | 0.343 |0.339|
>
> For more specifications refer to A.10.7 Performance for Item Recommendation Task.
>
> ------------------
> **W2: Is the method scalable, since the server has to receive FSS updates, perform secure aggregation, update the model, and perform inference all at once? It seems that there would be quite a bottleneck once the number of devices increase to realistic numbers.**
>
> Thanks for your comment. We would like to clarify the scalability of server cost for training and inference phases respectively. We also add the discussion in Appendix A.10.8 Server Computation Cost.
>
> - Training: The server computation cost to perform the secure aggregation scales linearly with the participating devices in each iteration. In reality, we typically select only a proportion of device (instead of all) to participate per iteration. The server cost is practical especially compared with the homomorphic encryption (HE) scheme in Table R2. We can see that the server cost for GREC is approximately 130x faster than the typical HE protocol on average.
>
> [**Table R2.** Server computation cost (in minutes) per iteration for ML1M dataset. ]
> |# of Active Users |100| 200 | 300 | 400 | 500 |
> |-|-|-|-|-|-|
> |HE (CKKS)| 127.26| 244.51 | 391.76 | 519.05  | 646.38|
> |GREC| 0.99| 1.92 | 2.93 | 3.88  | 4.90|
>
> - Inference: The server inference cost for GREC is the same as centralized setting, where the user data is uploaded to the server directly for recommendation.
>
> **Do note that the server computation cost is not the bottleneck in cross-device FL setting, as it typically holds strong computation power and they can use techniques such as parallelization to speed up the computation.** Instead, the bottleneck lies in the limited resource of user device, i.e., limited upload bandwidth and computation power. To address this bottleneck, we propose a efficient SecAgg protocol with reduced communication cost, and design a cloud-based inference framework using LDP post-processing techniques.

---

> ### Author Response · Authors · 2024-11-21
> **Rebuttal (Part-II)**
>
> **Q1: It would be nice if the authors could include the Related Works section within the main body.**
>
> Thanks for your suggestion. The Related Works section is deferred to appendix due to page limits. We would definitely include that to the main body if more pages are allowed.
>
> --------------
> **Q2: If server's collude, will FSS fail in terms of privacy? What would motivate two separate servers to participate in training and take on extra computational burdens? When would two separate servers be a realistic setting?**
>
> Thanks for your question. The privacy information would be leaked if the two servers collude. Actually, **two non-colluding parties is a common assumption in many multi-party computation (MPC) protocols [1-4]**. The user privacy is guaranteed as long as one of the two servers is honest. In practice, the two servers can be: (1) a cloud service provider who make the recommendation, and a third party who provides the cryptography or evaluation service; or (2) Two third parties who provide the cryptography or evaluation service. The service provider is motivated to employ other parties for the secure training by the following reasons: (1) strict data privacy regulation, and (2) business advantage by using privacy-preserving training.
>
> -------------------
> **Q3: For clarification, is m simply the number of items that can be recommended? I was curious when the inequality $m'<mbd/((\lambda+2)logm+bd)$ would fail to hold. Is this ever a concern?**
>
> Thanks for your question. m is the number of all items that can be recommended, and m' can be treated as the number of rated items per user. **The m' has to be a very large number to break the inequality.** In Table R3, we show the maximum value (or breakpoint) of m' where GREC gains an communication advantage over General-purpose SecAgg. It can be observed that the breakpoint of $m'$ is sufficiently large, over 50\% of the total item size $m$. It is highly improbable for a user to rate such a substantial proportion of items in practical scenarios.
>
> [**Table R3.** Maximum Value of $m'$ for Communication Cost Advantage over General-purpose SecAgg under Various Embedding Dimension $d$.]
> | |ML100K | ML1M | ML10M | ML25M | Yelp |
> |-|-|-|-|-|-|
> | |(1.7k Items)|(3.9k Items)|(10.7k Items)|(62.4k Items)|(93.4k Items)|
> |d=64|1001 | 2210 | 5775 | 31038 | 45418|
> |d=128|1255| 2817| 6408 | 37453 | 56031|
> |d=512|1550 | 3547| 9655 | 55418| 82495|
>
> ---------------------
> **Q4: Related works for efficient RecSys training.**
>
> Thanks for providing the relevant and inspiring works. We have included those papers in the related work section.

---

> ### Author Response · Authors · 2024-11-21
> **Rebuttal (Part-III)**
>
> [1] Corrigan-Gibbs, H., & Boneh, D. (2017). Prio: Private, robust, and scalable computation of aggregate statistics. In 14th USENIX symposium on networked systems design and implementation (NSDI 17) (pp. 259-282).
>
> [2] Addanki, S., Garbe, K., Jaffe, E., Ostrovsky, R., & Polychroniadou, A. (2022, September). Prio+: Privacy preserving aggregate statistics via boolean shares. In International Conference on Security and Cryptography for Networks (pp. 516-539). Cham: Springer International Publishing.
>
> [3] Boneh, D., Boyle, E., Corrigan-Gibbs, H., Gilboa, N., & Ishai, Y. (2021, May). Lightweight techniques for private heavy hitters. In 2021 IEEE Symposium on Security and Privacy (SP) (pp. 762-776). IEEE.
>
> [4] Mohassel, P., & Zhang, Y. (2017, May). Secureml: A system for scalable privacy-preserving machine learning. In 2017 IEEE symposium on security and privacy (SP) (pp. 19-38). IEEE.

---

> ### Author Response · Authors · 2024-11-25
> **Looking forward to hearing from you**
>
> Dear Reviewer cpnX,
>
> Thanks again for the recognition and valuable comments.
>
> We hope our responses have adequately addressed your previous concerns. With the discussion period drawing to a close, we look forward to hearing from you and would be happy to address any remaining concerns that you may still have.
>
> Thanks,
>
> Authors of Submission 8695

---

> > ### Author Response · Authors · 2024-12-02
> > **Look forward to hearing from you as reviewer reply ends in 24 hours**
> >
> > Dear Reviewer cpnX,
> >
> > We sincerely appreciate your time and effort to review our work. As the reviewer reply will end in less than 24 hours, we are eager to understand if our response has addressed your concerns. Any additional insights would be helpful to us.
> >
> > **Thank you once again for your valuable feedback and recognition of our work. You constructive feedback contributes to making our work more thorough and robust.** If you have any remaining questions, we would be glad to discuss and address them.
> >
> > Regards,
> >
> > Authors of Submission 8695

---

> > > ### Comment · Reviewer_cpnX · 2024-12-02
> > > **Late Reply**
> > >
> > > Dear Authors,
> > >
> > > I am terribly, terribly sorry for my late reply. Your responses have answered my questions. The empirical performance in reducing communication costs is impressive. While I think the paper is worthy of acceptance (given my score of acceptance), I am simply a bit unconfident in my overall assessment due to a lack of knowledge in DP and Secure Aggregation. I do think that the authors have done a great job during the rebuttal, and I want to reiterate that the paper is worthy of acceptance.

---

> > > > ### Author Response · Authors · 2024-12-03
> > > >
> > > > Dear Reviewer cpnX,
> > > >
> > > > Thank you very much for your kind reply and continued support of our work. We greatly appreciate your thoughtful feedback and are glad that our responses addressed your questions and concerns. Your comments have been invaluable in helping us improve the presentation and soundness of our work.
> > > >
> > > > Once again, thank you for your time, effort, and constructive feedback throughout this process.
> > > >
> > > > Best regards,
> > > >
> > > > Authors of Submission 8695

---

### Official Review · Reviewer_xYsi · 2024-11-05

**Soundness:** 3
**Presentation:** 4
**Contribution:** 3
**Rating:** 6
**Confidence:** 3

**Summary:**

This paper considers two key challenges in secure federated recommendation systems (FRS) for resource-constraint devices: minimizing communication costs and reducing computation/storage costs on edge devices. To address the first challenges, the proposed method, named GREC, introduces a functional secret sharing (FSS)-based Secure Aggregation (SecAgg) protocol that leverages the sparsity of item embeddings to improve communication efficiency. For the second challenge, GREC employs a user-side post-processing local differential privacy (LDP) mechanism during inference phase. This paper empirically demonstrates that GREC can reduce communication costs by up to 90x and user-side inference time by 11x compared to existing baselines.

**Strengths:**

- This paper considers a timely and important problem in FRS with privacy enhancements (secagg and DP).
- This paper use a funcional secret sharing (FSS) for secure aggregation on top of lerveraging the sparsity of item embeddings, which is both novel and practical. This allows GREC to significantly reduce the communication burden without sacrificing model performance or privacy.
- The introduction of a post-processing LDP mechanism that shifts much of the computation to the cloud, while maintaining privacy guarantees, is a practical and thoughtful contribution. The approach provides a good balance between privacy protection and computational/storage complexities.

**Weaknesses:**

- The comparison between GREC and general-purpose SecAgg in Section 3.1.3 and Table 1 may not be entirely fair, as only GREC takes advantage of the sparsity of item embeddings. Other works, such as Liu, Tao, et al. (2023) also leveraged sparsity to reduce communication costs of the secure aggregation protocol. For a fair assessment, GREC should be compared to such approaches both in asymptotic analysis and experiments.
  - Liu, Tao, et al. "Efficient and secure federated learning for financial applications." Applied Sciences 13.10 (2023): 5877.
- While the paper provides a detailed privacy analysis for the training phase, the analysis for the inference phase, especially in the context of cloud-based inference, could be more thorough. The paper assumes that the cloud environment is fully secure, but this assumption may not always hold in real-world deployments, which could compromise the overall privacy guarantees.

**Questions:**

- In the paper, the authors assume that $m'$ (the number of rated items) and $\mathcal{I}_u$ (the set of rated items for user $u$) remain fixed. Could the authors clarify how the system handles scenarios where a user rates new items, leading to changes in $\mathcal{I}_u$? Specifically, how does this affect the secure aggregation process and communication overhead?
- While the paper focuses on user-side privacy, a more detailed discussion on the assumptions about the security of cloud-based servers during inference would add depth to the privacy analysis. It would be beneficial to explore potential scenarios where the server could be compromised and how GREC might mitigate such risks.

---

> ### Author Response · Authors · 2024-11-21
> **Rebuttal (Part-I)**
>
> We thank the reviewer for the recognizing the value of our paper. Hope our response below could address the reviewer's concern.
>
> **W1 & Q1: For a fair assessment, GREC should be compared to such approaches that leverage sparsity to reduce communication costs of the secure aggregation protocol.**
>
> Thanks for your comment and suggestion. We would like to compare GREC with existing sparse aggregation protocols in terms of security and efficiency.
>
> For security, existing protocols [1][2] that leverage sparsity to reduce communication costs of SecAgg **fail to ensure that the server learns no information about individual updates except the aggregated gradients**.  Considering two representative frameworks, Secure Aggregation with Mask Sparsification (SecAggMask) [1] and Top-k Sparse Secure Aggregation (TopkSecAgg) [2]:
> - **Leakage of rated item index.** For SecAggMask, each user transmits the union of gradients & coordinates for non-zero updates and masks to the server. For TopkSecAgg, each user is required to upload the coordinate set of non-zero gradients along with a small portion of perturbed coordinates. In both methods, the server could narrow down the potential rated items to a much smaller set.
> - **Leakage of gradient values.** While TopkSecAgg protects the values of non-zero updates against the server, SecAggMask would reveal the plaintext values to the server. Specifically, SecAggMask randomly masks a portion of the gradients to reduce communication cost, and fails to ensure that all non-zero gradients would be masked against any attackers.
>
> In fact, achieving succinct communication cost (i.e., cost independent of or logarithmic in the total vector size) while ensuring that the server learns no information about individual vector is a highly non-trivially task, since the coordinates of the non-zero elements would be inevitably revealed to the server for existing sparse aggregation protocols. The location information is very sensitive for recommender system as they reveal the items rated by each user. **There is no work, to the best of our knowledge, that can achieve succinct communication cost while ensuring such security**.
>
> For efficiency comparison, we also present the communication cost on MF in Table R1. For full results and settings refer to Appendix A.10.6 Comparison with Sparse Aggregation Protocol. **Both sparse aggregation protocols result in higher communication cost than GREC** because: (1) Besides the non-zero embedding gradients, SecAggMask requires the user to send **a certain proportion of randomly masked zero updates** to the server. (2) To cancel out the mask values, in TopkSecAgg each user sends the **union** of rated item embeddings for all participating user, rather than the those for each single user.
>
> [**Table R1.** Communication cost (in MB) per user for GREC and Sparse SecAgg during upload transmission in one iteration using Matrix Factorization (MF).]
> | | |ML100K | ML1M | ML10M | ML25M | Yelp |
> |-|-|-|-|-|-|-|
> | | |(1.7k Items)|(3.9k Items)|(10.7k Items)|(62.4k Items)|(93.4k Items)|
> |MF|General SecAgg|0.87|2.02|5.55| 32.46|48.56|
> | |SecAggMask|0.27|0.61|1.66|9.60|14.35|
> | |TopkSecAgg|0.32|0.66|0.91|1.17|1.95|
> | |GREC|0.17|0.27|0.28|0.51|0.52|

---

> ### Author Response · Authors · 2024-11-21
> **Rebuttal (Part-II)**
>
> **W2: The paper assumes that the cloud environment is fully secure during inference, but this assumption may not always hold in real-world deployments, which could compromise the overall privacy guarantees.**
>
> Thanks for your comment. Our paper does not assume that the cloud server is fully secure to ensure inference privacy. As illustrated in Section 3.2.4 Privacy Analysis, **the user only transmits the privatized representation matrix to the server**, and all other private information is maintained secretly on their local device. Our algorithm ensures that the transmitted messages satisfy formal LDP. As the sensitive information never leaves user device, any outsider (including a compromised server) can obtain only the LDP privatized representation matrix and no other information about the user.
>
> ---------------------
> **Q1: In the paper, the authors assume that m' (the number of rated items) and $I_u$ (the set of rated items for user u) remain fixed. Could the authors clarify how the system handles scenarios where a user rates new items, leading to changes in $I_u$? Specifically, how does this affect the secure aggregation process and communication overhead?**
>
> Thanks for your question. Actually, m' is a pre-specified value unified for all users. It's pre-determined at the beginning of the protocol based on the average of all users' $m_u'$ (for more details refer to Appendix A.4 Standardization of Uploaded Item Size). Therefore, the average of $m_u'$ is only slightly impacted if merely a few users rate more items, and thus we could keep m' fixed in that case with almost no impact on the training process.
>
> If there are quite a number of users that rate more items during the training, then we can update the m' periodically during the training process to reflect the change. The communication cost would increase with larger m', but still have significant advantage compared with the General-purpose SecAgg. In Appendix A.10.9 Breakpoint Analysis of Communication Cost, m' should be increased to sufficiently large value, i.e., over 50\% of the total item size m, to break the advantage of GREC over general purpose SecAgg. It is highly improbable for a user to rate such a substantial proportion of items in practical scenarios.

---

> ### Author Response · Authors · 2024-11-21
> **Rebuttal (Part-III)**
>
> [1] Liu, T., Wang, Z., He, H., Shi, W., Lin, L., An, R., & Li, C. (2023). Efficient and secure federated learning for financial applications. Applied Sciences, 13(10), 5877.
>
> [2] Lu, S., Li, R., Liu, W., Guan, C., & Yang, X. (2023). Top-k sparsification with secure aggregation for privacy-preserving federated learning. Computers & Security, 124, 102993.

---

> ### Author Response · Authors · 2024-11-25
> **Looking forward to hearing from you**
>
> Dear Reviewer xYsi,
>
> Thanks again for the recognition and valuable comments.
>
> We hope our responses have adequately addressed your previous concerns. With the discussion period drawing to a close, we look forward to hearing from you and would be happy to address any remaining concerns that you may still have.
>
> Thanks,
>
> Authors of Submission 8695

---

> > ### Author Response · Authors · 2024-12-02
> > **Look forward to hearing from you as reviewer reply ends in 24 hours**
> >
> > Dear Reviewer xYsi,
> >
> > We sincerely appreciate your time and effort to review our work. As the reviewer reply will end in less than 24 hours, we are eager to understand if our response has addressed your concerns. Any additional insights would be helpful to us.
> >
> > **Thank you once again for your constructive feedback and support for our work. You insightful feedback helps make our work more thorough and robust.** If you have any remaining questions, we would be glad to discuss and address them.
> >
> > Regards,
> >
> > Authors of Submission 8695

---

### Official Review · Reviewer_WBjv · 2024-11-05

**Soundness:** 2
**Presentation:** 2
**Contribution:** 2
**Rating:** 5
**Confidence:** 4

**Summary:**

This paper presents a federated recommender system tailored for edge devices with limited computational and communication resources. The main technical contributions include a redesign for secure aggregation protocols, and a cloud inference approach with local differential privacy guarantee.

**Strengths:**

The paper proposes doubly efficient privacy-perserving recommender systems (GREC) consisting of both training
and inference phase. Both the SecAgg and LDP mechanisms are adapted thoughtfully to address specific limitations in edge environments.

**Weaknesses:**

1. While LDP is known to degrade model performance, GREC's method of user-side post-processing aims to minimize this. However, more detail on the denoising model’s training and its impact on utility in high-dimensional settings would strengthen the argument for its scalability and robustness.
2. While the GREC claims to have doubly efficiency approach, the relationship between its design for training and inference is not clear. The two parts seem to be independent.
3. For LDP based approach, the evaluation only uses epsilon = 1 setting, which is not enough. Usually for DP related work, tradeoffs between different privacy budgets (e.g., epsilon = 0.1, 0.3, 0.5, 0.7) and performance is expected.
4. Although the work's contributions to efficient and privacy-preserving federated learning are relevant to the ML community, its main technical novelty lies in the cryptographic components, rather than algorithm design/utility.

**Questions:**

1. Add detail on the impact on utility of LDP in high-dimensional settings is needed. That's when LDP would usually greatly degrade utility.
2. Explicitly discuss how the training and inference components interact or complement each other.
3. More evaluation regarding tradeoffs between different privacy budgets  (e.g., epsilon = 0.1, 0.3, 0.5, 0.7) and performance is expected.

---

> ### Author Response · Authors · 2024-11-21
> **Rebuttal (Part-I)**
>
> We thank the reviewer for the comment and suggestion. Hope our response below could address the reviewer's concern.
>
> **W1 & Q1: Add detail on the impact on utility of LDP in high-dimensional settings is needed.**
>
> Thanks for your comment and suggestion. We have added experiments for higher dimension up to 512. Table R1 presents the results for ML1M and Yelp using DeepFM. For full experiment results refer to Appendix A.10.4 Utility Analysis under High Dimension Setting. **The experiment results demonstrate the robustness of our post-processing LDP under high dimension settings.**
>
> [**Table R1.** Inference accuracy in terms of RMSE under various embedding dimensions.]
> | | |ML1M | | | Yelp| | |
> |-|-|-|-|-|-|-|-|
> |DeepFM|Dimension d| 64| 128 | 512 | 64| 128 | 512|
> | |Non-private |0.903| 0.901 | 0.901 |1.011| 1.019|1.003|
> | |Naive LDP| 2.275 | 2.342 | 2.345  |1.776 | 1.850|1.848|
> | |Post-processing LDP|0.905| 0.905| 0.903 |1.055| 1.054|1.056|
>
> ------------------
> **W2 & Q2: The relationship between GREC's design for training and inference.**
>
> Thanks for your comment. Our framework aims to **ensure consistent privacy and efficiency (in terms of user computation & communication cost) throughout the training and inference stage**.
> - **For privacy protection, the user embedding is maintained confidentially on user side throughout the training and inference phase.** In recommender system, the user embedding is highly personalized and thus would reveal sensitive information of the user. Such property motivates us to design a unified training and inference framework that hides the user embedding against the server. In training phase, the user embedding is updated locally on the client side. In the inference phase, the user embedding is privatized with LDP guarantee before being transmitted to the server.
> - **For efficiency, the item embedding is maintained on the server and never fully offloaded to the user device throughout the training and inference phase.** The item embedding dominates the total model size as the item size increases, and thus it is desirable to move the maintenance of the item embedding matrix on the server side. During training stage, the user keeps only the embeddings of the rated items to save computation power. In inference phase, there is no need for the user to store the item embedding matrix for private inference. Instead, they can send the privatized representation to the server and conduct subsequent post-processing locally with a lightweight denoise model.
> ------------------
> **W3 & Q3: For LDP based approach, the evaluation only uses epsilon = 1 setting, which is not enough. More evaluation regarding tradeoffs between different privacy budgets (e.g., epsilon = 0.1, 0.3, 0.5, 0.7) and performance is expected.**
>
> >For LDP based approach, the evaluation only uses epsilon = 1 setting, which is not enough.
>
> Prior ML studies typically use $\epsilon$ from 1 to 10 [1-8]. Following their work, we use privacy budget $\epsilon=1$. We also note that the $\epsilon$ used in the system deployed in real world ranges from 2 to 15 [9, 10]. For instance, Google's Gboard prediction uses the $\epsilon$ from 4 to 15 [9], and Apple's Sarifi uses $\epsilon$ from 4 to 8 [10].
>
> >More evaluation regarding tradeoffs between different privacy budgets (e.g., epsilon = 0.1, 0.3, 0.5, 0.7) and performance is expected.
>
> Thanks for your suggestion. We have added experiment results for privacy budget from 0.1 to 10, and have presented the performance in Table R2. For full experiment, please refer to Appendix A.10.5 Utility Analysis under Various Privacy Budgets. **While the accuracy for naive LDP degrades significantly as $\epsilon$ decreases to 0.1, the performance for our post-processing LDP remains robust to the change of privacy budget.**
>
> [**Table R1.** Inference accuracy in terms of RMSE under various arious privacy budgets.]
> | | |ML1M | | | Yelp| | |
> |-|-|-|-|-|-|-|-|
> |DeepFM|Privacy budget $\epsilon$| 0.1| 1 | 10 | 0.1| 1 | 10|
> | |Non-private |0.903|  |  |1.011| | |
> | |Naive LDP| 2.306 | 2.275 |1.979  |1.832 | 1.776|1.321|
> | |Post-processing LDP|0.911 | 0.907| 0.904 |1.054| 1.055|1.048|

---

> ### Author Response · Authors · 2024-11-21
> **Rebuttal (Part-II)**
>
> **W4: Although the work's contributions to efficient and privacy-preserving federated learning are relevant to the ML community, its main technical novelty lies in the cryptographic components, rather than algorithm design/utility.**
>
> Thanks for your comment. The contribution of our work lies in both machine learning and cryptography.
>
> Firstly, **we design a special-purpose SecAgg for certain ML task using cryptographic techniques**. Existing work on SecAgg protocols are almost general-purpose, while we propose a special-purpose SecAgg protocol tailoring to the properties of RecSys. In particular, we observe that the gradient for item embedding layer is a row-sparse matrix. Based on the characteristic, we design an efficient SecAgg algorithm that achieve succinct communication cost (i.e., cost independent of or logarithmic in the item size). This can be viewed as"ML for Crypto".
>
> Secondly, **our postprocessing LDP for secure inference uses ML approach for cryptogaphy tasks**. Existing tools for secure inference needs homomophic encryption or multiparty computation, incurring huge computation or communication overhead. To improve the efficiency over cryptographic-based secure inference protocol, we propose a post-processing LDP approach to conduct private inference using ML techniques.
> Unlike existing post-processing LDP approaches, our framework requires no additional efforts to obtain prior knowledge for user input, since the user embedding is obtained and kept locally on the edge device. This can be viewed as"ML for Crypto".

---

> > ### Comment · Reviewer_WBjv · 2024-12-02
> >
> > Thank you for the additional experimental results. I appreciate the efforts that you have made to improve your paper.
> >
> > For your response to W3, please note that it does not provide meaningful privacy protection when epsilon is above 1 according to the definition of differential privacy.
> >
> > While I still feel this paper should fits better for a security conference, I will raise my evaluation from 3 to 5.

---

> > > ### Author Response · Authors · 2024-12-02
> > >
> > > Dear Reviewer WBjv,
> > >
> > > Thank you for your thoughtful feedback and for raising your score. We are pleased that you recognize our effort to improve the paper.
> > >
> > > Regarding your comment on the use of $\epsilon$, we would like to clarify that we set $\epsilon = 1$ to demonstrate the robustness of our post-processing LDP method in a practical setting, where larger epsilon values (such as 2 to 10) are commonly used in real-world systems. But we also understand that stricter privacy guarantees ($\epsilon < 1$) are necessary in certain contexts, and thus we have conducted additional experiments with various $\epsilon$ (from 0.1 to 10) to show the trade-off.
> > >
> > > We believe that our work provides valuable insights for both privacy-preserving machine learning and cryptographic techniques in recommender systems, and we are confident that it will benefit the ML and security communities alike.
> > >
> > > Once again, we appreciate your constructive feedback and for raising your evaluation to 5. Your valuable suggestions have significantly contributed to improving the clarity and quality of our paper.
> > >
> > > Regards,
> > >
> > > Authors of Submission 8695

---

> ### Author Response · Authors · 2024-11-21
> **Rebuttal (Part-III)**
>
> [1] Yu, D., Naik, S., Backurs, A., Gopi, S., Inan, H. A., Kamath, G., ... & Zhang, H. Differentially Private Fine-tuning of Language Models. In International Conference on Learning Representations.
>
> [2] Papernot, N., Abadi, M., Erlingsson, Ú., Goodfellow, I., & Talwar, K. (2022, July). Semi-supervised Knowledge Transfer for Deep Learning from Private Training Data. In International Conference on Learning Representations.
>
> [3] Ding, Y., Wu, X., Luo, Y., Wang, H., & Pan, W. Delving into Differentially Private Transformer. In Forty-first International Conference on Machine Learning.
>
> [4] Li, X., Tramer, F., Liang, P., & Hashimoto, T. Large Language Models Can Be Strong Differentially Private Learners. In International Conference on Learning Representations.
>
> [5] Tramer, F., & Boneh, D. Differentially Private Learning Needs Better Features (or Much More Data). In International Conference on Learning Representations.
>
> [6] Xie, C., Lin, Z., Backurs, A., Gopi, S., Yu, D., Inan, H. A., ... & Yekhanin, S. Differentially Private Synthetic Data via Foundation Model APIs 2: Text. In Forty-first International Conference on Machine Learning.
>
> [7] Choquette-Choo, C. A., Dvijotham, K. D., Pillutla, K., Ganesh, A., Steinke, T., & Thakurta, A. G. Correlated Noise Provably Beats Independent Noise for Differentially Private Learning. In The Twelfth International Conference on Learning Representations.
>
> [8] Yu, Y., Sanjabi, M., Ma, Y., Chaudhuri, K., & Guo, C. ViP: A Differentially Private Foundation Model for Computer Vision. In Forty-first International Conference on Machine Learning.
>
> [9] https://arxiv.org/pdf/2305.18465
>
> [10] https://www.apple.com/privacy/docs/Differential_Privacy_Overview.pdf

---

> ### Author Response · Authors · 2024-11-25
> **Looking forward to hearing from you**
>
> Dear Reviewer WBjv,
>
> Thanks again for the valuable comments.
>
> We hope our responses have adequately addressed your previous concerns. With the discussion period drawing to a close,  we look forward to hearing from you and would be happy to address any remaining concerns that you may still have.
>
> Thanks,
>
> Authors of Submission 8695

---

### Official Review · Reviewer_qrKK · 2024-11-08

**Soundness:** 3
**Presentation:** 3
**Contribution:** 2
**Rating:** 5
**Confidence:** 2

**Summary:**

This paper proposes GREC, doubly efficient privacy-preserving recommender systems consisting of both training and inference phase. The goal is to improve federated recommender systems with the constraints of upload bandwidth and limited user computational power and storage. For the training phase, the authors design a lossless secure aggregation protocol based on functional secret sharing. For the inference phase, a user sider post-processing local differential privacy algorithm is proposed to ensure privacy. Experimental results show significant communication cost reduction of GREC compared with general purpose SecAgg, as well as user-side computation time reduction.

**Strengths:**

This paper proposes a framework to solve an important problem: reducing communication cost and user-side computation time for federated recommender systems. Related papers are cited and discussed. The optimization on the training phase leverages functional secret sharing scheme for the point function. For the inference phase, an LDP with post-processing mechanism is proposed to enable users to make their data private and send the data to the server to reduce computation cost on the user-side. Experimental results are shown to support the effectiveness of the proposed framework.

**Weaknesses:**

* I found the contribution of this paper incremental. The functional secret sharing scheme for the point function mechanism is from previously published papers. The LDP with user-side post-processing mechanism is also similar to previous work that is cited.
* It was stated in the introduction that there are other existing compression methods, but they "often result in non-negligible accuracy loss". However, there is no experimental comparison with these compression methods, but only with "general purpose" SecAgg methods.
* No error bars on the experimental results.

**Questions:**

* I'm a bit surprised that the inference accuracy on a single user's LDP data (without de-noise) is only ~40% higher than the non-privacy data. Usually a single user's data with LDP and reasonable (epsilon, delta) should have very large error.  Is it user-level or record level LDP?
* In section 4.1, it is mentioned that you "sample a portion of top users ranked in descending order by their number of rated items". Why not just sample the users uniformly at random to better represent the dataset?
* Have you compared your framework with other compression algorithms?

---

> ### Author Response · Authors · 2024-11-21
> **Rebuttal (Part-I)**
>
> We thank the reviewer for the comment and suggestion. Hope our response below could address the reviewer's concern.
>
> **W1: The contribution of this paper incremental.**
>
> Thanks for your comment. We would like to clarify our contribution for the training and inference stage.
>
> For *training stage*, we emphasize that our constribution is a **special-purpose SecAgg tailored to recsys task** based on function secret sharing, **not** constructing function secret sharing forpoint function. Note that **functional secret sharing (FSS) scheme doesn't in general imply an efficient SecAgg scheme**. Here, we observe that **the gradient for item embedding layer is fixed-dimensional row-sparse matrix**. The updated matrix for item embedding is sparse, and each non-zero update would occupy an entire row. A straightforward application of FSS on each non-zero element, i.e., applying FSS on each dimension and item, results in a communication cost of $O(m'db+m'd\lambda \log m)$ for the item embedding layer. This may exceed the cost for general-purpose SecAgg that has communication $O(mdb)$. **Our approach innovatively represents each non-zero embedding as an FSS key, reducing the communication cost to $O(m'db+m'\lambda \log m)$.** The reduction is more prominent under higher embedding dimension $d$.
>
> For *inference stage*, the user embedding is highly personalized and could reveal the sensitive information of user attributes. Therefore, it is necessary to maintain the user embedding secretly on the user device.
> We observe that there are a bunch of works focusing on post-processing DP [1,2,3]. The key component of post-processing LDP is to obtain the prior knowledge of input vector. The major difference between ours and theirs is that **our framework requires no additional efforts to obtain prior knowledge for user input, while such effort might lead to privacy issues**. A clever design of GREC is that the user embedding is obtained and kept locally on the user device, allowing the denoise model to be trained in a federated manner. There is no need to attempt to gather prior knowledge in GREC.
>
> In summary, our framework offers significant advancements in privacy and efficiency across the recommendation pipeline, addressing challenges that have not been fully resolved in existing literature.
>
> ---------------------
> **W2 & Q3: Comparison with other compression algorithms.**
>
> Thanks for your comment and suggestion. Our method differs fundamentally with the compression algorithms in that GREC is a SecAgg protocol. By SecAgg, it means that:
>
> (1) GREC ensures **security**, i.e., the server learns nothing about individual gradients except the aggregation. Other compression techniques allow the server to access the individual gradients.
>
> (2) GREC is **lossless** in principle, and thus is lossless in practice. Other compression techniques are lossy in principle, and we found that they have non-negligible utility loss in practice according to our experiment results. Actually, we have provided the comparison in the initial version of our paper in A.10.1 and gave the reference in Section 4.2. During rebuttal we add one more compression baseline, Ternary Quantization (TernQuant). For your convenience we present the results in Table R1.
>
> FedMF w/ SVD and CoLR [4] represent dimension reduction approaches, FedMF w/ Top-K [5] employs the Top-K sparsification technique, and TernQuant [6] employs gradient quantization method. We can see that **under similar communication cost, the performance is degraded on an average by 7.2\%, 16.3\%, 13.7\%, and 29.2\% for FedMF w/ SVD, CoLR, FedMF w/ Top-K, and TernQuant, respectively**.
>
> [**Table R1.** RMSE and reduction ratio for various message compression methods on ML10M and Yelp. Reduction ratio refers to the ratio of uplink communication cost before and after the application of the compression mechanism.]
> | | |GREC | FedMF w/ SVD | CoLR | FedMF w/ Top-K | TernQuant |
> |:-|:-|:-:|:-:|:-:|:-:|:-:|
> |ML10M|RMSE| 0.894 $\pm$ 0.004| 0.903 $\pm$ 0.002 | 0.931 $\pm$ 0.002 | 0.906 $\pm$ 0.003 | 1.631 $\pm$ 0.007|
> | |Reduction Ratio|19.25| 16.00 | 16.00 | 16.00| 16.00|
> |Yelp|RMSE| 1.353 $\pm$ 0.004| 1.563 $\pm$ 0.007 | 1.894 $\pm$ 0.011 |1.829 $\pm$ 0.005 | 1.587 $\pm$ 0.007|
> | |Reduction Ratio|91.18| 16.00 | 16.00 | 16.00| 16.00|

---

> ### Author Response · Authors · 2024-11-21
> **Rebuttal (Part-II)**
>
> **W3: No error bars on the experimental results.**
>
> Thanks for your comment and suggestion. In response, we have conducted additional experiments for four rounds and provided the standard deviation & error bars in Table 3 (see Section 4.3) and Figure 3-4 (see Section 4.4). In Table R2 we provide a same of our results with error bars, and for full results refer to Section 4.3 Inference Utility Analysis and Section 4.4 Computation Analysis. Note that there's no error bar for communication cost since the value is uniquely determined by the specified hyperparameters.
>
> [**Table R2.** Inference accuracy in terms of RMSE for ML1M.]
> | |Non-private | Naive LDP | Post-processing LDP | Diff (%)  |
> |:-|:-:|:-:|:-:|:-:|
> |MF| 0.903 $\pm$ 0.002| 1.637 $\pm$ 0.008 | 0.919 $\pm$ 0.001 | 43.47 |
> |NCF| 0.897 $\pm$ 0.006| 1.169 $\pm$ 0.068 | 0.915 $\pm$ 0.002 | 25.83 |
> |FM| 0.906 $\pm$ 0.000| 2.350 $\pm$ 0.018 | 0.908 $\pm$ 0.000 | 48.95 |
> |DeepFM| 0.902 $\pm$ 0.001| 2.281 $\pm$ 0.007 | 0.905 $\pm$ 0.003 | 55.52|
>
> ----------------
>
> **Q1: I'm a bit surprised that the inference accuracy on a single user's LDP data (without de-noise) is only ~40\% higher than the non-privacy data.  Is it user-level or record level LDP?**
>
> It is a user-level LDP. The naive LDP is over 45\% higher than non-private setting on average. **Actually, 45\% is a sufficiently huge error rate in recommender system, returning even worse performance than using the overall average rating as the prediction.** The inference error for naive LDP is not exceptionally high (e.g., over 90\%) for the following reasons:
>
> - We clip the norm of privitized user representation for fair comparison with our framework. The clipped embeddings still satisfy LDP according to the post-processing immunity. After clipping, the deviation between the privatized and raw embeddings is bounded within a reasonable range, and thus the prediction error wouldn't be very large.
> - The server could still utilize the item bias term for prediction, though the user embedding is noisy.
> - Also note that we use the RMSE for naive LDP as the denominator to compute the difference. If using the RMSE for non-private setting as denominator, the difference is around 90\% on average.
> -------------------------
> **Q2: In section 4.1, it is mentioned that you "sample a portion of top users ranked in descending order by their number of rated items". Why not just sample the users uniformly at random to better represent the dataset?**
>
> The original dataset is too sparse to efficiently learn representative embedding for each item & user. Even after we perform the aforementioned sampling, the density of the dataset is still 0.11\%, much lower than that of the four movielens dataset that ranges from 0.25\% to 6.30\%. In research, it is a common practice to filter out users with too few rated time for highly sparse dataset [7].

---

> ### Author Response · Authors · 2024-11-21
> **Rebuttal (Part-III)**
>
> [1] Wang, H., Sudalairaj, S., Henning, J., Greenewald, K., & Srivastava, A. (2024). Post-processing private synthetic data for improving utility on selected measures. Advances in Neural Information Processing Systems, 36.
>
> [2] Balle, B., & Wang, Y. X. (2018, July). Improving the gaussian mechanism for differential privacy: Analytical calibration and optimal denoising. In International Conference on Machine Learning (pp. 394-403). PMLR.
>
> [3] Mai, P., Yan, R., Huang, Z., Yang, Y., & Pang, Y. Split-and-Denoise: Protect large language model inference with local differential privacy. In Forty-first International Conference on Machine Learning.
>
> [4] Nguyen, N. H., Nguyen, T. A., Nguyen, T., Hoang, V. T., Le, D. D., & Wong, K. S. (2024, May). Towards Efficient Communication and Secure Federated Recommendation System via Low-rank Training. In Proceedings of the ACM on Web Conference 2024 (pp. 3940-3951).
>
> [5] Gupta, V., Choudhary, D., Tang, P., Wei, X., Wang, X., Huang, Y., ... & Mahoney, M. W. (2021, August). Training recommender systems at scale: Communication-efficient model and data parallelism. In Proceedings of the 27th ACM SIGKDD Conference on Knowledge Discovery & Data Mining (pp. 2928-2936).
>
> [6] Wen, W., Xu, C., Yan, F., Wu, C., Wang, Y., Chen, Y., & Li, H. (2017). Terngrad: Ternary gradients to reduce communication in distributed deep learning. Advances in neural information processing systems, 30.
>
> [7] Hariadi, A. I., & Nurjanah, D. (2017, November). Hybrid attribute and personality based recommender system for book recommendation. In 2017 International conference on data and software engineering (ICoDSE) (pp. 1-5). IEEE.

---

> ### Author Response · Authors · 2024-11-25
> **Looking forward to hearing from you**
>
> Dear Reviewer qrKK,
>
> Thanks again for the valuable comments.
>
> We hope our responses have adequately addressed your previous concerns. With the discussion period drawing to a close,  we look forward to hearing from you and would be happy to address any remaining concerns that you may still have.
>
> Thanks,
>
> Authors of Submission 8695

---

> > ### Comment · Reviewer_qrKK · 2024-11-25
> >
> > Thank you for the response! I have some follow up questions:
> >
> > * You mentioned "GREC ensures security, i.e., the server learns nothing about individual gradients except the aggregation. Other compression techniques allow the server to access the individual gradients." Is it true that there is no previous study on sparse secure aggregation?
> >
> >
> > * You mentioned "The major difference between ours and theirs is that our framework requires no additional efforts to obtain prior knowledge for user input, while such effort might lead to privacy issues."
> > My understand is that user side post-processing preserves privacy because these raw user data is never sent to the server. Would you elaboration on what "prior knowledge for user input" is used in "Split-and-Denoise: Protect large language model inference with local differential privacy." [3] that has privacy issues?

---

> > > ### Author Response · Authors · 2024-11-26
> > > **Reply to Follow up Questions (Part-I)**
> > >
> > > Thank you very much for taking time to review our rebuttal. Hope our responses below could answer your questions.
> > >
> > > **Follow-up Q1: You mentioned "GREC ensures security, i.e., the server learns nothing about individual gradients except the aggregation. Other compression techniques allow the server to access the individual gradients." Is it true that there is no previous study on sparse secure aggregation?**
> > >
> > > Thanks for your question. Compression techniques typically aim to reduce the size of transmitted gradients or models through lossy methods. For sparse secure aggregation, we think the reviewer refers to **secure aggregation protocols that utilize the model sparsity to reduce communication cost using lossless methods**.
> > >
> > > According to our response to Reviewer xYsi's **W1 & Q1**, existing sparse secure aggregations **fail to ensure that the server learns no information about individual updates except the aggregated gradients**. Leveraging the sparsity to reduce communication cost of SecAgg while ensuring security is a highly non-trivial task. To our best knowledge, there is no work that can achieve succinct communication cost (i.e., cost independent of or logarithmic in the total vector size) while ensuring that the server learns no information about individual vector. **A key challenge is that the index or coordinates of the non-zero elements would be inevitably revealed to the server for existing sparse aggregation protocols.** For instance, two representative sparse aggregation frameworks proposed by existing works, secure Aggregation with Mask Sparsification (SecAggMask) [1] and Top-k Sparse Secure Aggregation (TopkSecAgg) [2], fail to address this challenge:
> > >
> > > - **Leakage of rated item index.** For SecAggMask, each user transmits the union of gradients & indexes for non-zero updates and masks to the server. For TopkSecAgg, each user is required to upload the coordinate set of non-zero gradients along with a small portion of perturbed coordinates. In both methods, the server could **narrow down the potential rated items to a much smaller set**.
> > >
> > > - **Leakage of gradient values.** While TopkSecAgg protects the values of non-zero updates against the server, SecAggMask would reveal the plaintext values to the server. Specifically, SecAggMask randomly masks a portion of the gradients to reduce communication cost, and fails to ensure that all non-zero gradients would be masked against any attackers.
> > >
> > > In contrast, our framework ensures that the server learns no information about individual updates except the aggregated gradients. For more details refer to Appendix *A.10.6 Comparison with Sparse Aggregation Protocol*.
> > >
> > > [1] Liu, T., Wang, Z., He, H., Shi, W., Lin, L., An, R., & Li, C. (2023). Efficient and secure federated learning for financial applications. *Applied Sciences*, 13(10), 5877.
> > >
> > > [2] Lu, S., Li, R., Liu, W., Guan, C., & Yang, X. (2023). Top-k sparsification with secure aggregation for privacy-preserving federated learning. *Computers & Security*, 124, 102993.

---

> > > ### Author Response · Authors · 2024-11-26
> > > **Reply to Follow up Questions (Part-II)**
> > >
> > > **Follow-up Q2: You mentioned "The major difference between ours and theirs is that our framework requires no additional efforts to obtain prior knowledge for user input, while such effort might lead to privacy issues." My understand is that user side post-processing preserves privacy because these raw user data is never sent to the server. Would you elaboration on what "prior knowledge for user input" is used in "Split-and-Denoise: Protect large language model inference with local differential privacy." [3] that has privacy issues?**
> > >
> > > Thanks for your question. You are correct in that the raw user data are never sent to the server during both the training and inference stage. Obtaining prior knowledge for user input might lead to privacy issues.
> > >
> > > In SnD[3], the prior knowledge primarily refers to **the token embedding layer deployed on user side**. Both SnD and GREC operate on discrete input data—discrete tokens in SnD and user IDs in GREC. To process these, an embedding layer is required to transform the inputs into continuous representations (a token embedding layer in SnD and user embeddings in GREC). In SnD, **the model owner is required to share the token embedding layer as prior knowledge to encode user's input**. Therefore, the token embedding layer is public to both the users and server, which may cause two privacy issues:
> > >
> > > - For the case of language model, the model parameters are proprietary to the server. Revealing the token embedding layer to users could compromise server's privacy.
> > >
> > > - Adapting to FL-based RecSys, the alternative--user embedding--is highly personalized and could reveal user's sensitive information. Sharing that to the server could increase privacy leakage of the user.
> > >
> > > [3] Mai, P., Yan, R., Huang, Z., Yang, Y., & Pang, Y. Split-and-Denoise: Protect large language model inference with local differential privacy. In Forty-first International Conference on Machine Learning.
> > >
> > > --------------------
> > > We greatly appreciate your reply and inquiries. If you have additional questions, we would be glad to discuss and address them.

---

> ### Author Response · Authors · 2024-12-02
> **Look forward to hearing from you as reviewer reply ends in 24 hours**
>
> Dear Reviewer qrKK,
>
> We sincerely appreciate your time and effort to review our paper and rebuttal. As the reviewer reply will end in less than 24 hours, we are eager to understand if our response has addressed your concerns. Any additional insights would be helpful to us.
>
> **Thank you once again for your valuable feedback! Your thoughtful feedback contributes to enhancing our work.** If you have any remaining questions, we would be glad to discuss and address them.
>
> Regards,
>
> Authors of Submission 8695

---

> > ### Comment · Reviewer_qrKK · 2024-12-02
> >
> > Thank you for your detailed response! I have raised my score.

---

> > > ### Author Response · Authors · 2024-12-03
> > >
> > > Dear Reviewer qrKK,
> > >
> > > Thank you for your positive feedback and for raising your score. We are glad that our detailed responses addressed your concerns. **If there are any remaining concerns, we would be more than happy to provide further clarification.**
> > >
> > > Once again, thank you for your valuable feedback and for your thoughtful consideration of our submission.
> > >
> > > Best regards,
> > >
> > > Authors of Submission 8695

---

### Author Response · Authors · 2024-11-21
**Looking forward to further discussions to address concerns**

Dear Reviewers,


We sincerely appreciate your valuable comments and suggestions. We have uploaded an updated version of the manuscript, with revisions coloured in blue. We hope that our responses have adequately addressed your concerns. We look forward to hearing from you and shall be grateful for any feedback you give to us.


Best Regards,

Authors of Submission 8695

---

### Meta-Review · Area_Chair_36ML · 2024-12-21

**Metareview:**

The submission proposes GREC, a federated learning system consisting of (i) a training phase consisting of a lossless secure aggregation protocol based on functional secret sharing, and (ii) a user-side local differentially private mechanism that allows shifting the bulk of the computation to the cloud (instead of storing and maintaining the entire model locally, which is resource-intensive).

While the author(s) answer several of the questions of the reviewers during the rebuttal phase, the main concern remains that the contribution of the paper is too incremental to meet the bar for publication at ICLR.

**Additional Comments On Reviewer Discussion:**

The reviewers raised different concerns regarding this submission including:

1) the contribution of the paper is too incremental for ICLR
2) the paper might be a better a fit for a security conference
3) the epsilon values used in the evaluation
4) the prior literature and baselines
5) the experimental evaluation
6) the threat model

It is appreciated that the authors added evaluations for various epsilon values between 0.1 and 10, and that they satisfactorily answered the concerns related to 2), 3), 4), 5) and 6). However, the concern regarding 1) remains.

---

### Decision · Program_Chairs · 2025-01-22

Reject